# A Decade-Long Evaluation of Neonatal Septicaemic *Escherichia coli*: Clonal Lineages, Genomes, and New Delhi Metallo-Beta-Lactamase Variants

Amrita Bhattacharjee,[a] Kirsty Sands,[b,c] Shravani Mitra,[a] Ritojeet Basu,[d] Bijan Saha,[e] Olivier Clermont,[f,g] Shanta Dutta,[a] Sulagna Basu[a]

[a]Division of Bacteriology, ICMR-National Institute of Cholera and Enteric Diseases, Kolkata, West Bengal, India
[b]Division of Medical Microbiology, Institute of Infection and Immunity, Cardiff University, United Kingdom
[c]Ineos Oxford Institute of Antimicrobial Research, Department of Biology, University of Oxford, United Kingdom
[d]Department of Economics, University of Warwick, Coventry, United Kingdom
[e]Department of Neonatology, Institute of Post-Graduate Medical Education & Research and SSKM Hospital, Kolkata, West Bengal, India
[f]Université de Paris, IAME, UMR1137, INSERM, Paris, France
[g]Université Sorbonne Paris Nord, IAME, Paris, France

**ABSTRACT** Longitudinal studies of extraintestinal pathogenic *Escherichia coli* (ExPEC) and epidemic clones of *E. coli* in association with New Delhi metallo-$\beta$-lactamase ($bla_{NDM}$) in septicaemic neonates are rare. This study captured the diversity of 80 *E. coli* isolates collected from septicaemic neonates in terms of antibiotic susceptibility, resistome, phylogroups, sequence types (ST), virulome, plasmids, and integron types over a decade (2009 to 2019). Most of the isolates were multidrug-resistant and, 44% of them were carbapenem-resistant, primarily due to $bla_{NDM}$. NDM-1 was the sole NDM-variant present in conjugative IncFIA/FIB/FII replicons until 2013, and it was subsequently replaced by other variants, such as NDM-5/-7 found in IncX3/FII. A core genome analysis for $bla_{NDM}{}^{+ve}$ isolates showed the heterogeneity of the isolates. Fifty percent of the infections were caused by isolates of phylogroups B2 (34%), D (11.25%), and F (4%), whereas the other half were caused by phylogroups A (25%), B1 (11.25%), and C (14%). The isolates were further distributed in approximately 20 clonal complexes ($ST_C$), including five epidemic clones (ST131, ST167, ST410, ST648, and ST405). ST167 and ST131 (subclade H30Rx) were dominant, with most of the ST167 being $bla_{NDM}{}^{+ve}$ and $bla_{CTX-M-15}{}^{+ve}$. In contrast, the majority of ST131 isolates were $bla_{NDM}{}^{-ve}$ but $bla_{CTX-M-15}{}^{+ve}$, and they possessed more virulence determinants than did ST167. A single nucleotide polymorphism (SNP)-based comparative genome analysis of epidemic clones ST167 and ST131 in a global context revealed that the study isolates were present in close proximity but were distant from global isolates. The presence of antibiotic-resistant epidemic clones causing sepsis calls for a modification of the recommended antibiotics with which to treat neonatal sepsis.

**IMPORTANCE** Multidrug-resistant and virulent ExPEC causing sepsis in neonates is a challenge to neonatal health. The presence of enzymes, such as carbapenemases ($bla_{NDM}$) that hydrolyze most $\beta$-lactam antibiotic compounds, result in difficulties when treating neonates. The characterization of ExPECs collected over 10 years showed that 44% of ExPECs were carbapenem-resistant, possessing transmissible $bla_{NDM}$ genes. The isolates belonged to different phylogroups that are considered to be either commensals or virulent. The isolates were distributed in around 20 clonal complexes ($ST_C$), including two predominant epidemic clones (ST131 and ST167). ST167 possessed few virulence determinants but was $bla_{NDM}{}^{+ve}$. In contrast, ST131 harbored several virulence determinants but was $bla_{NDM}{}^{-ve}$. A comparison of the genomes of these epidemic clones in a global context revealed that the study isolates were present in close proximity but were distant from global isolates. The presence

Address correspondence to Sulagna Basu, basus@niced.gov.in, or supabasu@yahoo.co.in.

The authors declare no conflict of interest.

of epidemic clones in a vulnerable population with contrasting characteristics and the presence of resistance genes call for strict vigilance.

**KEYWORDS** *Escherichia coli*, phylogroup, epidemic clones, virulence, antibiotic resistance, New Delhi metallo-$\beta$-lactamases, India

*E*scherichia coli remains an important cause of neonatal sepsis in both developing and developed nations (1). Being a facultative anaerobe, *E. coli* is one of the first organisms to occupy the gut of neonates after birth (2). The bacteria that colonize the neonatal gut are generally transmitted from the mother to the child during birth or are acquired from the immediate environment (3). Most of the *E. coli* that colonize the gut are commensals that remain in the gastrointestinal tract, but some may translocate to the bloodstream and cause sepsis (4, 5). The immature gut barrier, the pristine gut, and the specialized traits of the bacteria may all contribute to the success of the translocation process (4, 5).

The difference between *E. coli* as a commensal versus disease-causing agent, at sites outside the intestine (extraintestinal pathogenic *E. coli* [ExPEC] isolates), is not always distinct, particularly in the neonate, as breaches in the gut barrier may allow commensal *E. coli* to cause disease (6–8). The existence of *E. coli* as a commensal (primarily in the gut but also in the environment) and as a successful pathogen that can cause sepsis, meningitis, urinary tract infection, osteomyelitis, wound-site infection, pyelonephritis, etc. testifies to its versatility (8). This versatility is observed in its genome and also in its function or potential to cause disease (9). Next-generation sequencing has led to a better and more in-depth understanding of the core and accessory genome. Based on the core genome, *E. coli* has distinct phylogenetic groups (A, B1, B2, C, D, E, F, G), of which B2 and, to a lesser extent, D, are considered to be ExPECs, whereas A and B1 are considered to be commensals (10).

A part of the accessory genome is acquired antimicrobial resistance genes (ARGs). Previously, it was believed that ARGs impose a "fitness cost" and that *E. coli* that were virulent therefore possessed fewer ARGs (10). The virulent strains (B2 phylogroup) possessed fewer antibiotic resistance genes than did the commensals (A, B1). However, a recent study has suggested that this may no longer be true (11). The epidemic clone ST131 has several antibiotic resistance determinants, and such strains are difficult to treat (12). Although several resistance genes are present in Gram-negative bacteria, one particular gene, namely, $bla_{NDM}$, which is a metallo-$\beta$-lactamase (MBL), has spread across the globe within a span of about 10 years (13–15). Several $bla_{NDM}$ variants have already been reported in Enterobacterales, including *E. coli* (16). In developing countries, higher rates of neonatal sepsis, combined with higher rates of antibiotic resistance, compared to the developed world, has created a situation in which the drugs recommended by the World Health Organization (WHO) for the treatment of sepsis are no longer effective in many cases (1, 17). The presence of MBLs in bacteria limits the therapeutic potential of antibiotics (18).

ExPECs have been well-studied, but there are considerably fewer studies that explore the ExPECs that cause neonatal infections/sepsis (19, 20). We explored a collection of ExPECs from the neonatal intensive care unit (NICU) over a span of 10 years in terms of phylogroup, sequence type (ST), international high-risk clones, presence, and transmission of $bla_{NDM}$ in an effort to understand any epidemiological pattern that is emerging across the years.

## RESULTS

**Identification of *Escherichia coli*.** 70 *E. coli* were identified from the blood of septicaemic neonates (2009 to 2019), and, furthermore, 10 *E. coli* isolates were identified from sources other than blood, such as endotracheal aspirate ($n = 2$), stool ($n = 2$), pus ($n = 2$), and peritoneal fluid ($n = 4$), not necessarily from the same neonates from whom blood was collected.

**Antibiotic susceptibility and prevalence of resistance determinants.** The isolates were resistant to second-generation (52/80, 65%) and third-generation cephalosporins (67/80, 84%), amikacin (39/80, 49%), gentamicin (53/80, 66%), ciprofloxacin (71/80, 89%), trimethoprim-sulfamethoxazole (62/80, 77%), aztreonam (67/80, 84%), and meropenem (35/80, 44%). The isolates were susceptible to only colistin and tigecycline. Thirty-five out of 80 isolates screened positive for the combination disk test, which indicated the production of MBL. Isolates that were positive in the combination disk test had higher minimum inhibitory concentration (MIC) values (MIC$_{50}$, 16 mg/L, MIC$_{90}$, 64 mg/L) except one (EN5076, 2 mg/L) (Table 1). The others had lower MIC values ($<$1 mg/L) for meropenem (MEM).

All of the isolates were categorized into two subpopulations for further analysis, based on the presence/absence of $bla_{NDM}$. The $bla_{NDM}$-$^{ve}$ isolates ($n = 45$) were comparatively more susceptible to other antibiotics, whereas the $bla_{NDM}$$^{+ve}$ isolates were highly resistant. There was a significant difference between the two groups in their nonsusceptibility to certain antibiotics, such as amikacin (*P* value $<$0.0001), gentamicin (*P* value $<$0.0001), and the second and third-generation cephalosporins (*P* values of $<$0.0001 and 0.0015, respectively) (Table 2).

Most isolates, whether harboring $bla_{NDM}$ or not, possessed several $\beta$-lactamases, such as $bla_{CTX-M}$, $bla_{TEM}$, $bla_{OXA-1}$, and $bla_{SHV}$ in percentages of 73%, 44%, 41%, and 28%, respectively. The most common metallo-$\beta$-lactamases found in the isolates was $bla_{NDM}$ (44%). Serine-$\beta$-lactamases such as $bla_{OXA-48-like}$ carbapenemase were present along with $bla_{NDM}$ in three isolates (EN5239, EN5378, EHN5379) (Table 1). Aminoglycoside-modifying enzymes, such as *rmtB* (34%), *armA* (13%), and *rmtC* (4%) were present. The efflux pump modulator gene *qepA* was present in a single isolate (EN5101) (Table 1) that coharbored $bla_{NDM}$. The fluoroquinolone resistance genes *aac-(6')-Ib-cr* (51%), *qnrB* (21%), *qnrS* (16%), *oqxA* (8%), and *oqxB* (11%) were also present. The carriage of multiple resistance determinants, such as $\beta$-lactamases, 16s rRNA methylases, plasmid-mediated quinolone resistance (PMQR), and AmpC, were higher in the $bla_{NDM}$$^{+ve}$ isolates (Table 2), compared to the $bla_{NDM}$-$^{ve}$ isolates. The distribution of different resistance determinants in the two subpopulations varied considerably (Table 2).

**A shift to NDM-5 from NDM-1 over time.** All of the carbapenem-resistant isolates ($n = 35$) were found to possess $bla_{NDM}$ as the major carbapenemase. A sequence analysis of the $bla_{NDM}$ revealed the presence of different NDM-variants, such as NDM-1 ($n = 12$), NDM-5 ($n = 19$), NDM-7 ($n = 3$), and NDM-15 ($n = 1$). The common mutation present in NDM-5, NDM-7, and NDM-15 was M154L, compared to NDM-1. Additional mutations (V88L for NDM-5, D130N for NDM-7, and A233V for NDM-15) were also present (15). The sequencing of $bla_{OXA-48-like}$ revealed that $bla_{OXA-181-like}$ carbapenemase was present along with $bla_{NDM-5}$ in the phylogroup A and ST$^{IP}$2 in Institut Pasteur scheme (ST$^{IP}$) isolates ($n = 3$). Two of these isolates (EN5378 and EN5379) were indistinguishable by PFGE (Table 1, Fig. S1).

NDM-1 was predominant until 2013, but, subsequently, other variants emerged. After 2014, NDM-1 was completely replaced with NDM-variants, such as NDM-5/NDM-7 (Fig. 1). The distribution of different NDM-variants, with their respective phylogroups and STs, across a period of 10 years is presented in Fig. 1.

**Transmissibility of metallo-$\beta$-lactamases and plasmid profiles.** Out of 35 $bla_{NDM}$$^{+ve}$ isolates, 30 transferred $bla_{NDM}$-harboring plasmids to *E. coli* J53. For the 5 isolates that could not successfully transfer the plasmid via conjugation, transformants (TFs) were analyzed. Additional resistance determinants, such as $bla_{CTX-M}$, $bla_{TEM}$, $bla_{CMY}$, *rmtB*, and *aac(6')-Ib-cr* were also found to cotransfer in different combinations with $bla_{NDM}$. A detailed characterization of the $bla_{NDM}$$^{+ve}$ *E. coli* isolates and their transconjugants (TCs) and TFs is provided in Table 1. The TCs and TFs exhibited variable ranges of MIC values for MEM: 1 mg/L to 64 mg/L (Table 1).

Several different plasmid scaffolds have been identified via PCR-based replicon typing (PBRT) and whole genome sequencing (WGS). Different incompatibility groups, such as A/C, L/M, FII, FIIK, FIIS, FIA, FIB, FIB-M, HIB-M, HI1$\alpha$, I1$\alpha$, I1$\gamma$, X1, X2, X3, U, Y, N, L, P, and R were found along with different types of Col plasmids, including Col

**TABLE 1** Characterization of $bla_{NDM}$[+ve] isolates with respect to the genetic environment, resistance determinants, plasmid types, and integrons[a]

| Isolates | Source | Isolation year | Phylogroup | ST[P] | MIC value (mg/L) | NDM variants | Genetic environment of $bla_{NDM}$ | Additional resistance determinants | Plasmid Inc types | Integrons |
|---|---|---|---|---|---|---|---|---|---|---|
| EN5054 | Blood | 2009 | C | 922 | 8 | NDM-1 | Truncated ISAba125 upstream and $ble_{MBL}$ downstream | $bla_{TEM}$, $bla_{OXA-1}$, $bla_{CMYγ}$ armA, aac(6')-Ib-cr | FIA, FIB, FII, I1γ, M | In46 |
| EN5054.TC | | | | | 4 | NDM-1 | | $bla_{TEM}$, $bla_{OXA-1}$, armA, aac(6')-Ib-cr | FII, M | |
| EN5076 | Blood | 2010 | C | 471 | 2 | NDM-1 | Truncated ISAba125 upstream and $ble_{MBL}$ downstream | $bla_{TEM}$, $bla_{CTX-M}$ rmtB | FIA, FIB, FII, FIIK, R | In27 |
| EN5076.TF | | | | | 8 | NDM-1 | | $bla_{TEM}$, $bla_{CTX-M}$ rmtB | FII | |
| EN5090 | Blood | 2011 | C | 922 | >32 | NDM-1 | Full-length ISAba125 upstream, $ble_{MBL}$ downstream | $bla_{TEM}$, $bla_{CTX-M}$, $bla_{CMYγ}$ rmtB | FII, I1γ, Y | In27 |
| EN5090.TC | | | | | 8 | NDM-1 | | $bla_{TEM}$, rmtB | FII | |
| EN5095 | Blood | 2011 | C | 471 | 32 | NDM-1 | Truncated ISAba125, IS5 upstream and $ble_{MBL}$ downstream | $bla_{CTX-M}$, rmtB, aac(6')-Ib-cr | FIB, FII | In27, In191 |
| EN5095.TC | | | | | 4 | NDM-1 | | rmtB | FII | |
| EN5101 | Blood | 2011 | A | 974 | 4 | NDM-1 | Full-length ISAba125 upstream, $ble_{MBL}$ downstream | $bla_{CTX-M}$, $bla_{TEM}$, $bla_{OXA-1}$, $bla_{CMYγ}$ rmtB, qnrB, qnrS, qepA, aac(6')-Ib | FIA, FIB, I1γ, R, X1 | In191 |
| EN5101.TF | | | | | 8 | NDM-1 | | $bla_{TEM}$, $bla_{OXA-1}$, $bla_{CTX-M}$ $bla_{CMYγ}$, rmtB, aac(6')-Ib-cr, qnrB, qepA | R | |
| NBE0002 | Stool | 2011 | C | 922 | 32 | NDM-1 | Full-length ISAba125 upstream, $ble_{MBL}$ downstream | $bla_{CTX-M}$, $bla_{TEM}$, $bla_{OXA-1}$, rmtB, aac(6')-Ib-cr, $bla_{CMY}$ | HI2, I1γ, FII | In27, In191, In705 |
| NBE0002.TC | | | | | 16 | NDM-1 | | $bla_{CTX-M}$, $bla_{TEM}$, $bla_{OXA-1}$, rmtB, aac(6')-Ib-cr | FII | |
| NBE0003 | Stool | 2011 | C | 922 | 8 | NDM-1 | Full-length ISAba125 upstream and $ble_{MBL}$ downstream | $bla_{CTX-M}$, $bla_{TEM}$, $bla_{OXA-1}$, $bla_{CMYγ}$ rmtB, aac(6')-Ib-cr | HI2, I1γ, FII | In27, In191, In705 |
| NBE0003.TC | | | | | 32 | NDM-1 | | $bla_{CTX-M}$, $bla_{TEM}$, $bla_{OXA-1}$, rmtB, aac(6')-Ib-cr | FII, I1γ | |
| NBE0004 | Endotracheal aspirate | 2012 | A | 2 | 64 | NDM-7 | Insertion of ISKpn26 in full-length ISAba125 upstream, $ble_{MBL}$ downstream | $bla_{CTX-M}$, $bla_{OXA-1}$, rmtB, qnrS, aac(6')-Ib-cr | FIA, FIK, FII, HIB-M, N | In54, In2101[b] |
| NBE0004.TC | | | | | 32 | NDM-7 | | $bla_{OXA-1}$, aac(6')-Ib-cr | FII, FIA | |
| EN5132 | Blood | 2013 | B2 | 43 | 4 | NDM-1 | Full-length ISAba125 upstream, $ble_{MBL}$ downstream | $bla_{CTX-M}$, $bla_{TEM}$ $bla_{OXA-1}$, armA, aac(6')-Ib-cr | C, FIB, FII, N | In54 |
| EN5132.TC | | | | | 2 | NDM-1 | | $bla_{CTX-M}$, $bla_{TEM}$, $bla_{OXA-1}$, armA, aac(6')-Ib-cr | C | |
| NBE0005 | Pus | 2013 | B2 | 43 | 8 | NDM-1 | Truncated ISAba125 upstream and $ble_{MBL}$ downstream | $bla_{CTX-M}$, $bla_{TEM}$, armA, aac(6')-Ib, aac(6')-Ib-cr, $bla_{DHA}$ | C, FIB, FII | ND |

**TABLE 1** (Continued)

| Isolates | Source | Isolation year | Phylogroup | ST[IP] | MIC value (mg/L) | NDM-variants | Genetic environment of $bla_{NDM}$ | Additional resistance determinants | Plasmid Inc types | Integrons |
|---|---|---|---|---|---|---|---|---|---|---|
| NBE0005.TC | | | | | 8 | NDM-1 | | $bla_{CTX-M}$, $bla_{OXA-1}$, $bla_{DHA}$, aac(6')-Ib, aac(6')-Ib-cr, | C, FII | In54, In191 |
| EN5134 | Blood | 2013 | A | 2 | 24 | NDM-15 | Full-length IS$Aba$125 upstream, $ble_{MBL}$ downstream | $bla_{CTX-M}$, $bla_{TEM}$, $bla_{OXA-1}$, rmtB, aac(6')-Ib-cr | FIA, FII, I1α | ND |
| EN5134.TC | | | | | 2 | NDM-15 | | $bla_{TEM}$, rmtB | FII | |
| EN5141 | Blood | 2013 | B2 | 43 | 4 | NDM-1 | Truncated IS$Aba$125 upstream and $ble_{MBL}$ downstream | $bla_{CTX-M}$, $bla_{TEM}$, $bla_{OXA-1}$, armA, aac(6')-Ib-cr | FIB, FII, HIB-M | |
| EN5141.TC | | | | | 1.5 | NDM-1 | | $bla_{TEM}$ | HIB-M | |
| EN5143 | Blood | 2013 | C | 471 | 4 | NDM-5 | Full-length IS$Aba$125 upstream, $ble_{MBL}$ downstream | $bla_{TEM}$, rmtB | FIA, FII, I1γ | In27 |
| EN5143.TC | | | | | 1.5 | NDM-5 | | $bla_{TEM}$, rmtB | FIA, FII | |
| NBE0006 | Pus | 2013 | A | 43 | 16 | NDM-5 | Truncated IS$Aba$125 upstream and $ble_{MBL}$ downstream | $bla_{TEM}$, $bla_{CMY}$, rmtB | Y, FII | In27 |
| NBE0006.TC | | | | | 8 | NDM-5 | | $bla_{TEM}$, $bla_{CMY}$, rmtB, | FII | |
| NBE0007 | Peritoneal Fluid | 2013 | A | 2 | 64 | NDM-5 | Truncated IS$Aba$125 upstream and $ble_{MBL}$ downstream | $bla_{TEM}$, $bla_{CMY}$, rmtB | Y, FII | In27 |
| NBE0007.TC | | | | | 8 | NDM-5 | | $bla_{TEM}$, $bla_{CMY}$, rmtB | FII | |
| NBE0008 | Peritoneal Fluid | 2013 | B2 | 43 | 8 | NDM-1 | Truncated IS$Aba$125 upstream and $ble_{MBL}$ downstream | $bla_{CTX-M}$, $bla_{TEM}$, $bla_{OXA-1}$, $bla_{DHA}$, armA, aac(6')-Ib-cr, | FIB, FII, FIIK, HIB-M | In54 |
| NBE0008.TC | | | | | 2 | NDM-1 | | $bla_{CTX-M}$, $bla_{TEM}$, $bla_{OXA-1}$, aac(6')-Ib-cr | FII | |
| EN5169 | Blood | 2014 | B1 | 635 | >32 | NDM-7 | IS5 upstream and $ble_{MBL}$ downstream | $bla_{TEM}$, rmtB | FIA, FIB, FII, I1γ, R, X3, X4, | In54 |
| EN5169.TC | | | | | 1 | NDM-7 | | $bla_{TEM}$ | I1γ, X3, X4 | |
| NBE0009 | Peritoneal Fluid | 2014 | A | 2 | 64 | NDM-5 | Truncated IS$Aba$125 upstream and $ble_{MBL}$ downstream | $bla_{TEM}$, $bla_{DHA}$, rmtB | FII, I1γ, Y | In27 |
| NBE0009.TC | | | | | 8 | NDM-5 | | rmtB | FII | |
| NBE0010 | Peritoneal Fluid | 2014 | C | 66 | 128 | NDM-1 | Full-length IS$Aba$125 upstream, $ble_{MBL}$ downstream | $bla_{CTX-M}$, $bla_{TEM}$, rmtB, qnrB, aac(6')-Ib | FII, HI-1A, HI-1B, X3 | In27, In54 |
| NBE0010.TC | | | | | 8 | NDM-1 | | $bla_{CTX-M}$, rmtB | FII, X3 | |
| EN5177 | Blood | 2014 | A | 2 | 4 | NDM-5 | Truncated IS$Aba$125 upstream and $ble_{MBL}$ downstream | $bla_{TEM}$, rmtB | FIB, FII | In27 |
| EN5177.TC | | | | | 1 | NDM-5 | | $bla_{TEM}$, rmtB | FIB, FII | |
| EN5192 | Blood | 2014 | D | 477 | >32 | NDM-5 | Truncated IS$Aba$125 upstream and $ble_{MBL}$ downstream | $bla_{CTX-M}$, $bla_{TEM}$, $bla_{OXA-1}$, rmtB, aac(6')-Ib-cr | FIA, FIB, FII | In27, In54 |
| EN5192.TC | | | | | 4 | NDM-5 | | $bla_{TEM}$ | FII | |
| EN5197 | Blood | 2014 | B1 | 58 | >32 | NDM-7 | Full-length IS$Aba$125 upstream, $ble_{MBL}$ downstream | $bla_{CTX-M}$, $bla_{TEM}$, $bla_{DHA}$, rmtB, armA, aac(6')-Ib-cr | FIA, FIB, FII, HI1, I1α, X3 | In27 |

**TABLE 1** (Continued)

| Isolates | Source | Isolation year | Phylogroup | ST$^{IP}$ | MIC value (mg/L) | NDM-variants | Genetic environment of $bla_{NDM}$ | Additional resistance determinants | Plasmid Inc types | Integrons |
|---|---|---|---|---|---|---|---|---|---|---|
| EN5197.TF | | | | | 32 | NDM-7 | | $bla_{CTX-M}$, $bla_{TEM}$ $bla_{DHA}$, $rmtB$ $aac(6')$-$Ib$-$cr$ | HI1, X3 | |
| EN5239 | Blood | 2015 | A | 2 | 8 | NDM-5 | Full-length IS$Aba125$ upstream, $ble_{MBL}$ downstream | $bla_{CTX-M}$, $bla_{TEM}$, $bla_{OXA-1}$, $bla_{OXA-181}$, $bla_{CMY}$ $qnrS$, $aac(6')$-$Ib$-$cr$ | FIA, FIB, FII | In27, In54 |
| EN5239.TF | | | | | 4 | NDM-5 | | $bla_{OXA-1}$, $bla_{CMY}$, $qnrS$ | FIA, FIB | |
| EN5286 | Blood | 2016 | F | ST$^{W}$648 | >32 | NDM-5 | Truncated IS$Aba125$ upstream and $ble_{MBL}$ downstream | $bla_{TEM}$, $bla_{SHV}$, $bla_{OXA-1}$, $qnrB$, $armA$, $aac(6')$-$Ib$-$cr$ | FIB, FIA, FII, HI2 | In27, In191, In705 |
| EN5286.TC | | | | | 2 | NDM-5 | | $bla_{TEM}$ | FII | |
| EN5308 | Blood | 2016 | A | 2 | >32 | NDM-5 | Full-length IS$Aba125$ upstream, $ble_{MBL}$ downstream | $bla_{CTX-M}$, $bla_{TEM}$, $bla_{OXA-1}$, $qnrS$, $armA$, $rmtB$, $aac(6')$-$Ib$-$cr$ | X1, FII | In27 |
| EN5308.TC | | | | | 64 | NDM-5 | | $bla_{CTX-M}$, $bla_{TEM}$, $bla_{OXA-1}$, $armA$, $aac(6')$-$Ib$-$cr$ | FII | |
| EN5313 | Blood | 2016 | A | 2 | >32 | NDM-5 | Full-length IS$Aba125$ upstream, $ble_{MBL}$ downstream | $bla_{CTX-M}$, $bla_{OXA-1}$, $qnrS$, $aac(6')$-$Ib$, $aac (6')$-$Ib$-$cr$ | FIIK, FII | ND |
| EN5313.TC | | | | | 0.5 | NDM-5 | | $bla_{OXA-1}$, $qnrS$, $aac (6')$-$Ib$-$cr$ | FIIK | |
| EN5317 | Blood | 2016 | B1 | 479 | >32 | NDM-5 | Truncated IS$Aba125$ upstream and $ble_{MBL}$ downstream | $bla_{CTX-M}$, $bla_{TEM}$, $rmtB$, $qnrB$ | HI2, I1α, FIB, FIA, I1γ, X2, FII | In27 |
| EN5317.TC | | | | | 16 | NDM-5 | | $bla_{TEM}$, $rmtB$ | FII, I1α | |
| EN5349 | Blood | 2017 | A | 2 | 128 | NDM-5 | Full-length IS$Aba125$ upstream, $ble_{MBL}$ downstream | $bla_{CTX-M}$, $bla_{TEM}$ $bla_{CMY}$ $rmtB$ $qnrS$ | FIA, FII, Y | In27, In406 |
| EN5349.TF | | | | | 32 | NDM-5 | | $bla_{CTX-M}$, $bla_{TEM}$ $bla_{CMY}$, $rmtB$, $qnrS$ | FII, Y | |
| EN5356 | Blood | 2017 | C | 471 | 4 | NDM-5 | Truncated IS$Aba125$ upstream and $ble_{MBL}$ downstream | $bla_{TEM}$, $bla_{SHV}$, $armA$, $rmtB$, $qnrB$, $qnrS$, $oqxB$ | FIB-M, FII, HIB-M I1γ | In27 |
| EN5356.TC | | | | | 32 | NDM-5 | | $bla_{TEM}$, $rmtB$ | FII | |
| EN5358 | Blood | 2017 | A | 2 | >8 | NDM-5 | Full-length IS$Aba125$ upstream, $ble_{MBL}$ downstream | $bla_{CTX-M}$, $bla_{SHV}$, $bla_{TEM}$, $bla_{OXA-1}$, $bla_{CMY}$, $armA$, $rmtB$, $qnrB$, $qnrS$, $oqxB$ | I1γ, HIB-M, Y, FIB-M, FII | In27 |
| EN5358.TC | | | | | 64 | NDM-5 | | $bla_{TEM}$, $bla_{OXA-1}$ | FII, I1γ | |
| EN5374 | Blood | 2018 | A | 2 | ≥16 | NDM-5 | Full-length IS$Aba125$ upstream, $ble_{MBL}$ downstream | $bla_{CTX-M}$ | FIB, FIA, FIIK, FII | In27 |
| EN5374.TC | | | | | 4 | NDM-5 | | $bla_{CTX-M}$ | FII, FIIK | |
| EN5378 | Blood | 2018 | A | 1135 | 8 | NDM-5 | Truncated IS$Aba125$ upstream and $ble_{MBL}$ downstream | $bla_{CTX-M}$, $bla_{OXA-181}$, $bla_{CMY}$, $rmtB$ | FIB, FIA, X3, I1γ, FII | ND |
| EN5378.TC | | | | | 32 | NDM-5 | | $bla_{CMY}$, $rmtB$ | FIA, FIB, FII | |
| EN5379 | Blood | 2019 | A | 1135 | 8 | NDM-5 | Truncated IS$Aba125$ upstream and $ble_{MBL}$ downstream | $bla_{CTX-M}$, $bla_{TEM}$, $bla_{OXA-181}$, $bla_{CMY}$, $rmtB$, $qnrS$ | FIB, FIA, X3, I1γ, FII | In27, In54 |
| EN5379.TC | | | | | 64 | NDM-5 | | $rmtB$, $qnrS$ | FIA, FIB, FII | |

**TABLE 1** (Continued)

| Isolates | Source | Isolation year | Phylogroup | ST[a] | MIC value (mg/L) | NDM-variants | Genetic environment of $bla_{NDM}$ | Additional resistance determinants | Plasmid Inc types | Integrons |
|---|---|---|---|---|---|---|---|---|---|---|
| EN5381 | Blood | 2019 | A | 2 | 8 | NDM-5 | Truncated *ISAba125* upstream and $ble_{MBL}$ downstream | $bla_{CTX-M}$, $rmtB$, $rmtC$, $oqxB$ | FIB, FIA | ND |
| EN5381.TC | | | | | 4 | NDM-5 | | $bla_{CTX-M}$, $rmtB$, $oqxB$ | FIA | |
| EN5383 | Blood | 2019 | A | 2 | 8 | NDM-5 | Truncated *ISAba125* upstream and $ble_{MBL}$ downstream | $bla_{CTX-M}$, $bla_{TEM}$, $bla_{SHV}$, $bla_{OXA-1}$, $rmtB$, $rmtC$, $oqxA$, $oqxB$, $aac$ $(6')$-$Ib$-$cr$ | FIA, FII, I1γ | In27, In191 |
| EN5383.TC | | | | | 0.5 | NDM-5 | | $bla_{CTX-M}$, $bla_{TEM}$, $bla_{OXA-1}$, $rmtB$, | I1γ | |

[a]TC, transconjugant; TF, transformant; ND, not determinable because WGS was not carried out for these isolates.
[b]Novel integron found in this study isolate.

**TABLE 2** Comparison of phylogroups, STs, antibiotic susceptibility, resistance, and virulence determinants between $bla_{NDM}{}^{+ve}$ and $bla_{NDM}{}^{-ve}$ isolates

| Attributes | Name | NDM-positive ($bla_{NDM}{}^{+ve}$) ($n = 35$) | | NDM-negative ($bla_{NDM}{}^{-ve}$) ($n = 45$) | |
|---|---|---|---|---|---|
| | | Frequency | Sequence type (ST$^{IP}$) | Frequency | Sequence type (ST$^{IP}$) |
| Phylogroup distribution with associated ST | A | 17 | 2, 974, 1135 | 3 | 2, 664 |
| | B1 | 3 | 58, 479, 635 | 6 | 21,58, 294, 369 |
| | B2 | 4 | 43 | 23 | 4, 6, 33, 36, 43, 53,129, 146, 506, 686 |
| | C | 9 | 66, 471, 922 | 2 | 471 |
| | D | 1 | 477 | 8 | 3, 8, 477, 678, 1134 |
| | F | 1 | 648$^c$ | 2 | 648$^c$ |
| | NA | | | 1 | 816 |
| Proportion of nonsusceptibility$^a$ | Piperacillin | 100% | | 91% | |
| | 2$^{nd}$ gen Cephalosporin$^b$ | 100% | | 38% | |
| | 3$^{rd}$ gen Cephalosporin$^b$ | 100% | | 71% | |
| | Ciprofloxacin | 100% | | 80% | |
| | Amikacin | 94% | | 13% | |
| | Gentamicin | 94% | | 44% | |
| | Aztreonam | 97% | | 73% | |
| | Trimethoprim/ Sulfamethoxazole | 94% | | 64% | |
| | Meropenem | 100% | | 0% | |
| | Colistin | 0% | | 0% | |
| | Tigecycline | 0% | | 0% | |
| Prevalence of genetic determinants | β-lactamases | $bla_{CTX-M}$ (71%), $bla_{TEM}$ (60%), $bla_{SHV}$ (31%), $bla_{OXA-1}$ (51%), | | $bla_{CTX-M}$ (75%), $bla_{TEM}$ (29%), $bla_{SHV}$ (24%), $bla_{OXA-1}$ (29%), | |
| | AmpC | $bla_{CMY}$ (27%), $bla_{DHA}$ (3%) | | $bla_{CMY}$ (9%), $bla_{DHA}$ (5%) | |
| | 16S rRNA methylase | $armA$ (26%), $rmtB$ (68%), $rmtC$ (8%) | | - | |
| | PMQR | $qnrB$ (26%), $qnrS$ (26%), $oqxA$(3%), $oqxB$ (14%) | | $qnrB$ (15%), $qnrS$ (7%), $oqxA$(13%), $oqxB$ (8%) | |
| | Aminoglycoside modifying enzyme | $aac$-(6′)-$Ib$ (11%), $aac$-(6′)-$Ib$-$cr$ (51%) | | $aac$-(6′)-$Ib$ (4%), $aac$-(6′)-$Ib$-$cr$ (51%) | |
| | Carbapenemases | $bla_{OXA-48}$ (7%) | | - | |
| Putative virulence determinants | afa | 6% | | 7% | |
| | fimH | 31% | | 76% | |
| | papA | 23% | | 44% | |
| | papC | 26% | | 55% | |
| | papG | 20% | | 29% | |
| | IronE.c. | 6% | | 22% | |
| | iucC | 31% | | 67% | |
| | cdtB | 0% | | 0% | |
| | cnf | 3% | | 24% | |
| | sfa | 3% | | 13% | |
| | traT | 94% | | 80% | |
| | ibeA | 0% | | 2% | |
| | usp | 17% | | 51% | |
| | iha | 9% | | 38% | |
| | PAI | 23% | | 40% | |
| | fyuA | 80% | | 64% | |
| | hlyA | 3% | | 27% | |
| | MTII | 23% | | 42% | |
| | cvaC | 0% | | 0% | |

$^a$The results of antibiotic nonsusceptibility were interpreted by following the guidelines of the European Committee on Antimicrobial Susceptibility Testing (EUCAST) for colistin and tigecycline and the Clinical and Laboratory Standard Institute (CLSI) 2020 for the rest of the antibiotics. *E. coli* ATCC 25922 was used as the quality control strain.
$^b$Disk diffusion data of cefoxitin (2nd Gen), cefotaxime (3rd Gen) (2009 to 2017), cefuroxime (2nd Gen), and ceftriaxone (3rd Gen) in Vitek 2 AST 280 (2018 to 2019).
$^c$Sequence types (ST) in the Warwick scheme (ST$^W$), as the ST could not be detected for 3 isolates in the Institut Pasteur scheme due to lack of *uidA* gene.

(pHAD28), Col(BS512), Col(MG828), Col440I, ColVc, Col156, and ColRNAI. The carriage of $bla_{NDM}$ was probably via the IncF group (FIA, FIB, FII, and FIIS) as most TCs harbored these plasmids. The carriage of $bla_{NDM}$ was mostly via the IncF group (FIA, FIB, FII, and FIIS) and, to a lesser extent, via IncX3, X4, R, A/C, Y, as reflected by the conjugation results. In addition, the typing of the F-plasmid (pMLST) using the WGS data suggested that F2:A-:B and F36:A4:B (FAB formula) were the common plasmid types associated with $bla_{NDM-1}$ and $bla_{NDM-5}$, respectively (Table 3).

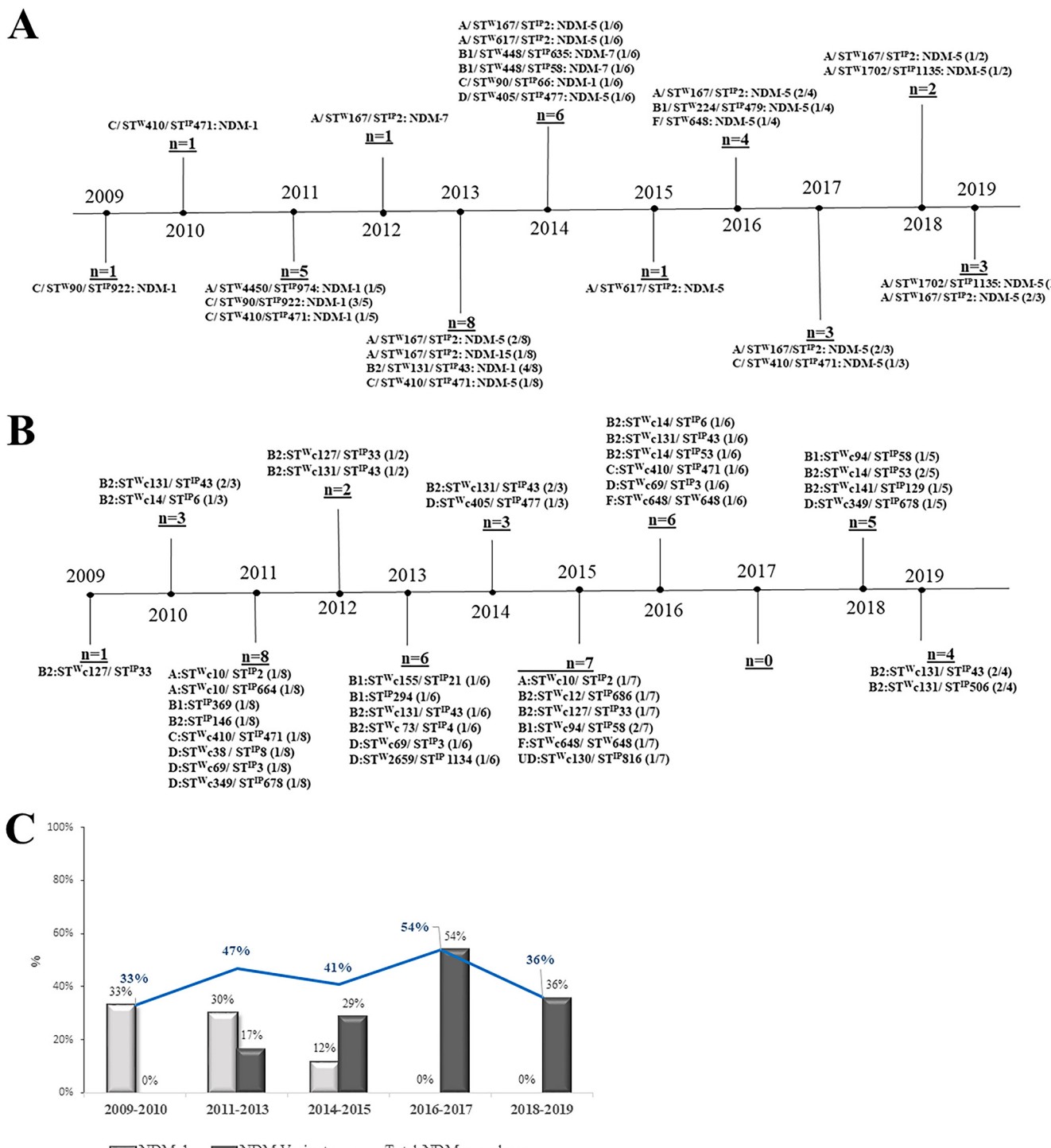

**FIG 1** The timeline of $bla_{NDM}^{+ve}$ (A) and $bla_{NDM}^{-ve}$ (B) isolates with phylogroups, sequence types (STs), and prevalence (*n*). For the $bla_{NDM}^{+ve}$ isolates both the Warwick scheme ($ST^W$) and the Institut Pasteur scheme ($ST^{IP}$) are mentioned with the $bla_{NDM}$-variants (A). For the $bla_{NDM}^{-ve}$ isolates, the phylogroups and the clonal complex in the Warwick scheme ($STc^W$) with $ST^{IP}$ are mentioned (B). The prevalence of NDM-1, NDM-5, and NDM-7 over 10 years (C).

**Genetic environment of $bla_{NDM}$ and integrons.** Both the PCR and WGS data were considered for the study of the genetic environment of the $bla_{NDM}^{+ve}$ isolates. At the upstream, $bla_{NDM}$ was always associated with IS*Aba*125, either in full-length (15/35) or truncated (18/35). Two isolates (EN5095 and EN5169) possessed IS5 with a remnant of IS*Aba125* within close proximity of $bla_{NDM}$. In another isolate (NBE0004), the insertion of IS*Kpn*26 between IS*Aba*125 and $bla_{NDM}$ was found. $ble_{MBL}$ was always found to be

**TABLE 3** Distribution of NDM-variants among different phylogroups, sequence types, serotypes, fimH-types, C-H types, and IncF types[a]

| NDM-Variants | Phylogroup (n) | ST$^W$ (n) | St$^{IP}$ (n) | Serotype (n) | fimH type | C-H type | IncF type FAB formula (n) |
|---|---|---|---|---|---|---|---|
| NDM-1 (10/30) | A (1) | 4450 | 974 | O53:H18 | H24 | 8-24 | K1:A13:B- |
| | B2 (2) | 131 | 43 | O25:H4 | H30 | 40-30 | F29:A-:B10 |
| | C (7) | 90 (5) | 922 (4) | O8:H9 | H142 | 4-142 | F31:A4: B1 (1); F2:A-:B- (3) |
| | | | 66 (1) | | | | F2:A-:B- (1) |
| | | 410 (2) | 471 (2) | O8:H9 | H24 | 4-24 | F36:A-:B32 |
| | | | | O78:H18 | | | F2:A4:B1 |
| NDM-5 (16/30) | A (11) | 167 (8) | 2 | O101:H9 (6) | Absent | 11-0 | F2:A-:B- (3); F36:A4:B (2); F31:A4:B1 (1) |
| | | | | O101:H5 (1) | | | F36:A4:B- (1) |
| | | | | O101:H33 (1) | | | F36:A4:B1 (1) |
| | | 617 (2) | | O101:H10 | | | F2:A-:B1 (1); F31:A24:B1 (1) |
| | | 1702 (1) | 1135 | O101:H9 | | | F2:A1:B49 |
| | B1 (1) | 224 | 479 | O8:H23 | H61 | 4-61 | F2:A-:B- |
| | C (2) | 410 | 471 | O0:H21 | H24 | 4-24 | F2:A1:B- (1) |
| | | | | O0:H9 | | | F44:A4:B1 (1) |
| | D (1) | 405 | 477 | O102:H6 | H27 | 37-27 | F2:A1:B49 |
| | F (1) | 648 | ND | O1:H6 | Absent | 4-0 | F2:A1:B1 |
| NDM-7 (3/30) | A (1) | 167 | 2 | O101:H9 | Absent | 11-0 | F2:A4:B36 |
| | B1 (2) | 448 (2) | 635 | O30:H8 | H35 | 6-35 | F2:A1:B49 (1) |
| | | | 58 | O4:H8 | | | F48:A1:B49 (1) |
| NDM-15 (1/30) | A | 167 | 2 | O101:H9 | Absent | 11-0 | F36:A4:B- |

[a]ST$^W$, sequence type in the Warwick scheme; ST$^{IP}$, sequence type in the Institut Pasteur scheme; *n*, number of isolates; C-H type, *fumC-fimH* typing; pMLST, plasmid multilocus sequence typing, ND, not determinable.

present downstream of all of the *bla*$_{NDM}$-producing isolates. Four different downstream combinations were found in the individual isolates in this region (Fig. 2).

For the integrons, In27 was frequently observed (*n* = 21), and it contained different genes (*dfrA12* [dihydrofolate reductase], *gcuF* [a hypothetical protein], *aadA2* [aminoglycoside 3'-adenyltransferase]) in their variable regions and conferred resistance against trimethoprim and streptomycin. In addition, In54 and In191 were identified in 8 and 7 isolates, respectively. A new class 1 integron, namely, In2101 (*estX-3*, which is a conserved hypothetical protein) was detected in a single isolate (NBE0004), according to the INTEGRALL database (http://integrall.bio.ua.pt/). 12 isolates were found to be carrying more than one class 1 integron (Table 1).

**Genetic relatedness of *bla*$_{NDM}$$^{+ve}$ isolates.** PFGE was carried out in the *bla*$_{NDM}$$^{+ve}$ isolates (*n* = 35), and it revealed that isolates were clonally diverse with four clonal clusters that were found across four different years: cluster 1 (EN5132, NBE0005, and

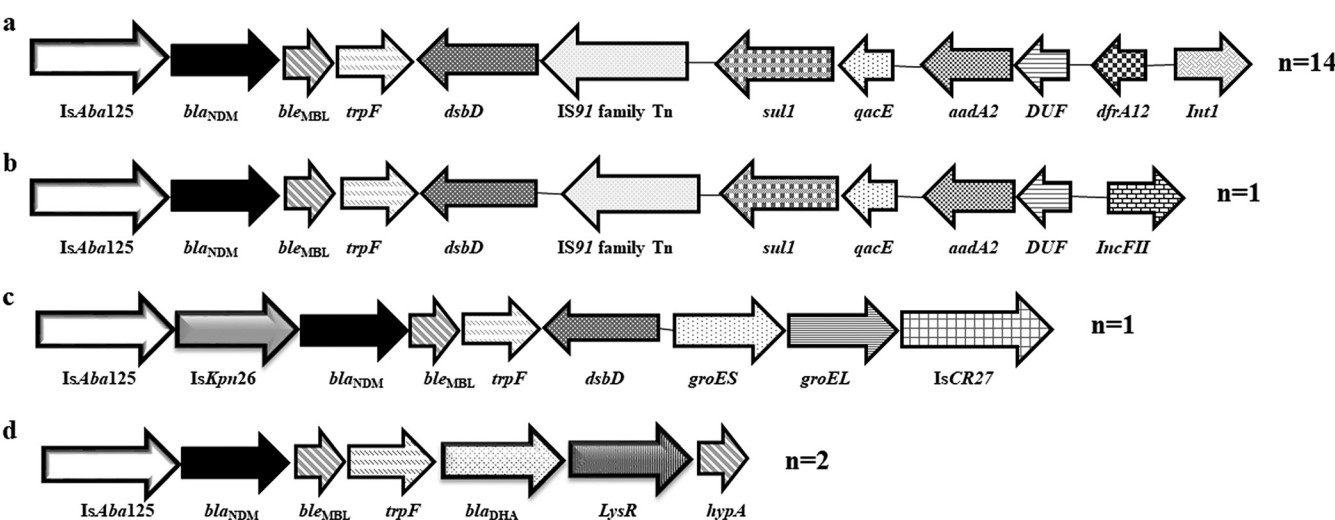

**FIG 2** The schematic diagram of the regions surrounding *bla*$_{NDM}$ in ExPEC isolates. Combination a for EN5076, EN5090, NBE0002, NBE0003, EN5143, NBE0006, NBE0007, EN5177, EN5197, EN5286, EN5308, EN5317, EN5349, and EN5358; combination b for EN5095; combination c for NBE0004; and combination d for EN5132 and NBE0008.

Tree scale: 0.01 ⊢————·————⊣

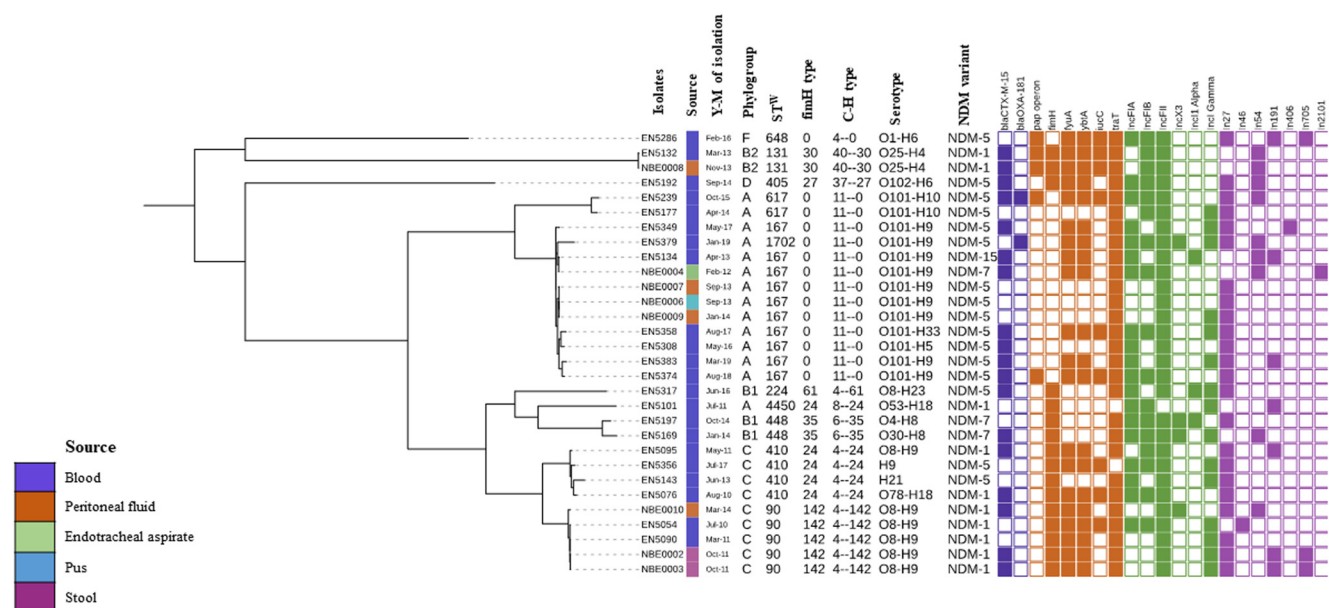

**FIG 3** The core genome phylogeny of the *bla*$_{NDM}$$^{+ve}$ study isolates. 30 out of 35 nonclonal *bla*$_{NDM}$$^{+ve}$ isolates are incorporated in this analysis. A colored box follows the dendrogram to indicate the source of each isolate. The year-month (Y-M) of sample isolation, phylogroup, ST$^W$ (Warwick scheme), fimH type, C-H (*fumC-fimH*) type, and serotype are mentioned for all isolates. The heat map indicates the presence/absence of resistance determinants (purple), virulence determinants (orange), plasmid replicon types (light green), and integrons (violet) that follow the NDM-variants.

EN5141; ST$^W$131, in 2013), cluster 2 (EN5308 and EN5313; ST$^W$167 in 2016), cluster 3 (EN5378 and EN5379; ST$^W$1702 in 2018), and cluster 4 (EN5381 and EN5383; ST$^W$167 in 2019) (Fig. S1). Based on the PFGE result, WGS was performed for isolates (*n* = 30) that included all of the distinct isolates and one representative from each different cluster (Fig. 3). The core genome phylogenetic tree (Fig. 3) also highlights the interspecies diversity among the *bla*$_{NDM}$$^{+ve}$ isolates.

The further analysis of the genomes showed different genetic characteristics, based on phylogroups, STs, serotypes, *fimH* types, C-H types, and NDM-variants (Table 3). The details of the NDM-variants, phylogroups, and STs are discussed separately in the respective sections. The different variants of NDM possessed different serotypes. It was noted that the serotypes of the *bla*$_{NDM-1}$$^{+ve}$ isolates were not identified in the other variants of *bla*$_{NDM-5/-7/-15}$. The *bla*$_{NDM}$$^{+ve}$ ExPECs were categorized into 15 different serotypes, and among them, the *bla*$_{NDM-1}$$^{+ve}$ isolates belonged to four serotypes, including O53:H18, O25:H9, O8:H9, and O78:H18; the *bla*$_{NDM-5}$$^{+ve}$ isolates belonged to O101:H9, O101:H5, and O101:H33; the *bla*$_{NDM-7}$$^{+ve}$ isolates belonged to O101:H9, O30:H8, and O4:H8; and the *bla*$_{NDM-15}$ isolates belonged to O101:H9 (Table 3). Six variants of Type 1 fimbrin D-mannose specific adhesin (*fimH*) were distributed in the *bla*$_{NDM}$-possessing isolates (Table 3).

**Phylogroups and sequence types (STs).** All of the isolates (blood and non-blood) belonged to six different phylogenetic groups, including A, B1, B2, C, D, and F. The majority of the strains belonged to phylogroup B2 (34%), and this was followed by A (25%), C (14%), D (11.25%), B1 (11.25%), and F (4%). For a single isolate, the phylogroup was undetected. An analysis of the blood isolates showed a distribution of B2 (38%), A (21%), D (13%), B1 (13%), C (11%). and F (4%). For isolates that were collected from different body sites, other than blood (*n* = 10), the phylogroup distribution was as follows: A (50%), C (30%), and B2 (20%). Phylogroup B2 isolates were identified each year, and they persisted throughout the 10 years. There was an increasing prevalence of phylogroup A over time (23% to 35% from 2011 to 2019). No such trend was noticed for the other phylogroups (Table 2; Fig. 1).

The sequence types of all isolates (blood and non-blood) were analyzed by the Institut Pasteur scheme (ST$^{IP}$), and their respective clonal complexes were detected using the goEBURST algorithm. The ST$^{IP}$s were also converted to the corresponding clonal complexes in the Warwick scheme (ST$^{W}$), as this scheme is easy to compare with global strains (Table S1). In addition, the $bla_{NDM}$$^{+ve}$ isolates were subjected to WGS, and the ST$^{W}$s for these isolates were determined from the WGS data.

All 80 isolates (with or without $bla_{NDM}$) were classified into 29 different ST$^{IP}$s: ST$^{IP}$2, 3, 4, 6, 8, 21, 33, 36, 43, 53, 58, 66, 129, 146, 294, 369, 471, 477, 479, 506, 635, 664, 678, 686, 816, and 922 as well as the three novel ST$^{IP}$974, 1134, and 1135. ST$^{IP}$2 (20%) was predominant, and it was followed by ST$^{IP}$43 (16%), ST$^{IP}$471 (7.5%), ST$^{IP}$58 (5%), ST$^{IP}$3 (3.75%); ST$^{IP}$53 (3.75%), and other lone ST$^{IP}$s. The predominant clone ST$^{IP}$2 was associated with phylogroup A. Other ST$^{IP}$s, such as ST$^{IP}$43, ST$^{IP}$58, and ST$^{IP}$471, were members of phylogroups B2, B1, and C, respectively. The isolates of phylogroup B2 were further divided into five subgroups: subgroups I (ST$^{W}$c131), III (ST$^{W}$c127), IV (ST$^{W}$c141), VI (ST$^{W}$c12), and VII (ST$^{W}$c14) (Table S1). ST$^{IP}$2, which was the most predominant ST$^{IP}$, corresponds to the epidemic clones ST$^{W}$167 and ST$^{W}$617. Similarly, ST$^{IP}$43, ST$^{IP}$471, and ST$^{IP}$477 correspond to ST$^{W}$131, ST$^{W}$410, and ST$^{W}$405, respectively, which are also globally disseminated high-risk clones. All ST$^{IP}$s with their respective clonal complexes (in ST$^{IP}$ and ST$^{W}$) are mentioned in (Table S1).

The novel ST$^{IP}$974 was found in a single isolate (EN5101) (Table 1) belonging to phylogroup A and possessed $bla_{NDM-1}$ (Table 1). The $bla_{NDM}$$^{+ve}$ isolates belonged to 11 ST$^{IP}$s: ST$^{IP}$2, 43, 58, 66, 471, 477, 479, 635, 922, 974 and 1135, which correspond to ST$^{W}$167 & 617, 131, 448, 90, 410, 405, 224, 448, 90, 4450 and 1702, respectively.

During the last few years (2016 to 2019) of the study period, ST$^{W}$167 was the major epidemic clone that was significantly associated with $bla_{NDM-5}$ (two-tailed *P* value of <0.0001) (Fig. 1A), whereas the other epidemic clone, namely, ST$^{W}$131, was mostly associated with the $bla_{NDM}$$^{-ve}$ (Fig. 1B) isolates that were collected until 2013 (the association with $bla_{NDM}$ was not found to be significant; two-tailed *P* value of 0.3351). ST$^{W}$410 persisted throughout the years with a low prevalence, and diversity in serotypes were observed. Two of them were $bla_{NDM-1}$$^{+ve}$ with two different serotypes (O78: H18 and O8:H9), whereas the other two isolates were $bla_{NDM-5}$$^{+ve}$ with somatic antigen O-negative.

**Virulence factors.** On the basis of the studied putative virulence determinants (*n* = 19), the most predominant VFs were *traT* (86.25%) and *fyuA* (71.25%), and these were followed by *fimH* (56.25%), *iucC* (51.25%), *papC* (42.5%), *usp* (36.25%), *papA* (35%), MTII (33.75%), *PAI* (32.5%), *papG* (25%), *iha* (25%), *hlyA* (16.25%), $iron_{E.c}$ (15%), *sfa/foc* (9%), and *afa* (6.25%), with *ibeA* being found only in a single isolate and with *cdtB* and *cvaC* being completely absent (Table 2). Phylogroup B2 was associated with the highest number of VFs (median VF score of 10), compared to other the phylogroups: phylogroup C (median VF score of 4), D (median VF score of 4), A (median VF score of 2) and B1 (median VF score of 2), where the median score is defined as the number of VFs carried by the 50% of the isolates.

**ST$^{W}$131 relatedness analysis.** The ST$^{W}$131 study isolates were subdivided into two clades, based on *fimH*-typing (clade A: H41 [*n* = 1] and clade C: H30R [*n* = 13]), and two subclades [H30R (1/13), H30Rx (11/13), and C1-M27 (1/13)]. The isolates were of two serotypes, namely, O16:H5 and O25:H4, for clades A and C, respectively. All of the isolates harbored $bla_{CTX-M-15}$ with chromosomal mutations in *gyrA* (DNA gyrase; S83L, D87N), *parC* (topoisomerase IV; S80I, E84V), and *parE* (topoisomerase IV; I529L) that conferred elevated fluoroquinolone resistance. Only one isolate (EN5115) possessed $bla_{CTX-M-27}$ belonging to the C1-M27 subclade. A specific group of resistance genes, including *aadA5*, *dfrA12*, *mdf(A)*, *mph(A)*, *sul1*, *tetA*, *aac-6'-lb-cr*, and $bla_{CTX-M-15}$, was found in almost all of the isolates. None of the isolates possessed $bla_{NDM}$, except for four (EN5132, EN5141, NBE0005, and NBE0008) that were found to be positive for $bla_{NDM-1}$ from the H30Rx subclade and to be carrying some additional resistance determinants (Fig. S2). Virulence determinants, such as heme uptake (*chuA*), curli fibers (*csgBDFG*), type 1 fimbriae (*fimABCDEFGHI*), yersiniabactin siderophore (*fyuA*, *irp1*, *irp2*, *iucABCD*, *iutA*, *ybtAEPQST*), outer membrane protein A (*ompA*), pap

operon (*papBCDFGH*), and others (*entABCDEFS*, *fepABCDGS*), were present in ST$^W$131. *papGII* was present in all of the H30Rx isolates. The pathogenicity-associated island (PAi) mostly encoded the *papGII* that was found in 50% of the isolates. The plasmid types, as revealed by pMLST, were as follows: F2:A1:B-, F29:A-:B10, and F36:A1:B20 and F1:A1:B16 for the H30Rx and F1:A2:B20 for the C1-M27 subclade.

For the ST$^W$131 contextual SNP analysis, 48 genomes from NCBI were included. The NCBI genomes were largely blood isolates, although some ($n = 11$) were urine isolates. In total, ST$^W$131 genomes from 10 countries were included, and the most distant isolate (SNP analysis against the reference EN5382, internal isolate) was a single isolate from the USA, as indicated by it having the longest branch length. The SNP phylogenetic tree is formed by two main branches with several subbranches. Isolate EN5382 (acting as the reference) was the only isolate from this study in the top branch (Fig. 4A), consisting of 11 isolates in total, and it was approximately 500 SNPs distant from the other branches.

Of the two main branches, there was an observed pattern in the carriage of the extended-spectrum $\beta$-lactamases (ESBL), especially CTX-M. The top branch was largely *bla*$_{\text{CTX-M}}$$^{\textbf{-ve}}$, whereas the majority of the isolates in the larger branch, including a subbranch, were *bla*$_{\text{CTX-M}}$$^{\textbf{+ve}}$. One branch of the ST$^W$131 isolates was exclusively *bla*$_{\text{CTX-M-15}}$$^{\textbf{+ve}}$, whereas a subbranch of the ST$^W$131 isolates carried *bla*$_{\text{CTX-M-27}}$. All of the isolates within this study, irrespective of their placement in the phylogenetic tree, were *bla*$_{\text{CTX-M-15}}$$^{\textbf{+ve}}$ (Fig. 4A). Genomes of the global isolates downloaded from NCBI were mostly *bla*$_{\text{NDM}}$$^{\textbf{-ve}}$, except for four from this study that possessed *bla*$_{\text{NDM-1}}$. WGS was carried out for two of these (EN5132, NBE0008), as the two other strains were clonal with EN5132 within this study. The isolates EN5132 and NBE0008 were within 20 SNPs and were collected during 2013, within a period of 3 months, whereas the isolate EN5179 sits on a separate branch and is 39 SNPs distant from the pair but is *bla*$_{\text{NDM}}$$^{\textbf{-ve}}$ (Fig. 3A). Additionally, two isolates within this study were closely related, being 8 pairwise SNPs distant: isolates EN5393 and EN5394 (Fig. 4A).

**ST$^W$167 relatedness analysis.** The high-risk clone ST$^W$167 ($n = 12$) isolates revealed some common patterns of resistance and virulence determinants. For the resistance determinants, *bla*$_{\text{NDM-5/-7/-15}}$, *aadA2*, *dfrA12*, *mph(A)*, *sul1*, and chromosomal mutations in *gyrA* (S23L, D87N), *parC* (S80I), and *parE* (S48A) were common in all of the strains. In addition to PCR data, WGS helped to explore more VF determinants, such as common pilus (*ecpABCDE*), lamin binding fimbriae (*elfACDH*), type I fimbriae (*fimDFG*), type IV pilli (*pilw*), invasins (*ibeBC*), yersiniabactin siderophore (*fyuA*, *irp1*, *irp2*, *ybtAEPQST*), *Klebsiella* sp. specific lipopolysaccharide (*rfb* locus), heat resistance (*hra*), increased serum survival (*iss*), and serum resistance-associated outer membrane protein (*traT*), which were present in all of the isolates. A serotype-based characterization showed that all of the isolates shared the same O (somatic, O101)-antigen biosynthesis gene cluster (21). In addition, a remnant of the *wca* operon, which is involved in the colanic acid biosynthesis in the LPS biosynthesis pathway, was present in all of the isolates. ST$^W$167 and its single-locus variant (SLV), such as ST$^W$617 and ST$^W$1702, did not possess any *fimH* types.

12 out of 35 *bla*$_{\text{NDM}}$$^{\textbf{+ve}}$ isolates belonged to ST$^W$167; however, a SNP analysis on WGS data further revealed genetic divergence within the ST$^W$167 isolates (Fig. 4B). A contextual SNP analysis, including $n = 33$ NCBI genomes from 4 continents with *bla*$_{\text{NDM}}$, indicated that all of the isolates were within a distance of approximately 500 SNPs. Further analysis revealed that seven isolates (including five from NCBI and two from this study) were single-locus variants of ST$^W$167, and this is denoted on the tree (Fig. 4B). The SNP phylogeny indicates that the blood isolates from this study were diverse, as is highlighted by the multiple branches. The isolates NBE0006, NBE0007, and NBE0009 are within 15 pairwise SNPs, and they were collected within 2013, whereas all of the other ST$^W$167 isolates within this study were >50 pairwise SNPs distant from each other (Fig. 4B).

**Association of clinical factors and neonate mortality.** Table 4 depicts the demographic, clinical, and bacterial data distribution among the discharged and expired

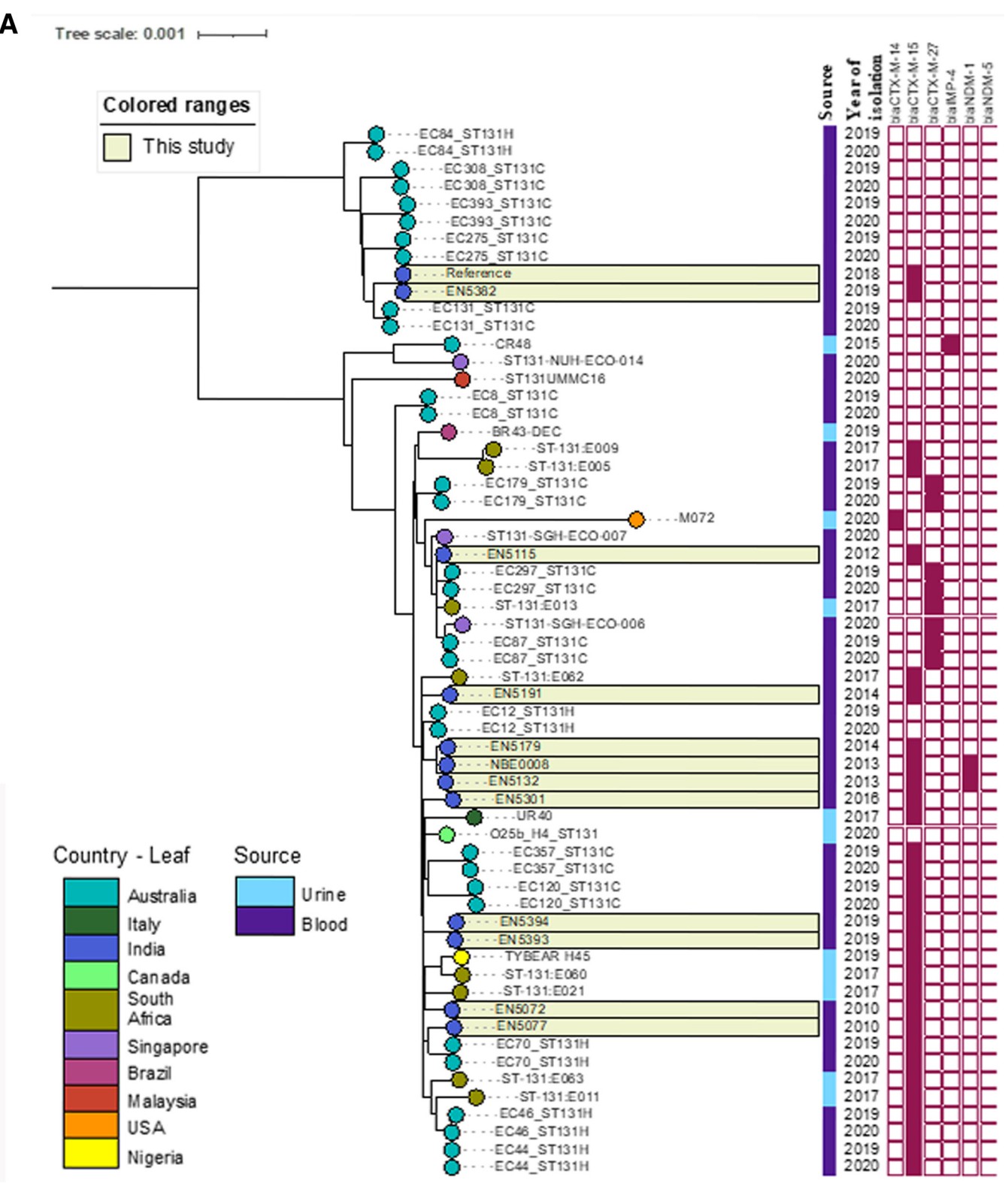

**FIG 4** The SNP-based core genome phylogeny of ST^W131 (A) and ST^W167 (B), including both study and global isolates. The isolates are color-coded at the endpoint by nation. The isolate sample collection year has been externally incorporated into the tree phylogeny along with other resistance determinants. The study isolates are highlighted with a light yellow color. (A) For the ST^W131 contextual SNP analysis, genomes from NCBI (*n* = 48) and the study isolates (*n* = 11) were included. The NCBI genomes were primarily blood isolates, although some (*n* = 11) were urine isolates. In total, ST^W131 genomes from 10 countries were included, and a SNP analysis was carried out against the reference EN5382. The presence/absence of $bla_{CTX-M-14/15/27}$, $bla_{IMP-4}$, and $bla_{NDM-1/-5}$ are indicated by a heat map (dark red). (B) For the ST^W167 contextual SNP analysis, genomes from NCBI (*n* = 33) and the study isolates (*n* = 14) were included. The NCBI genomes were isolated from blood, urine and rectal swabs, and they were collected from 9 different countries. The presence/absence of $bla_{NDM-1/-4/-5/-7/-15}$, $bla_{KPC-2}$, and $bla_{OXA-181}$ are indicated by heat map (dark red).

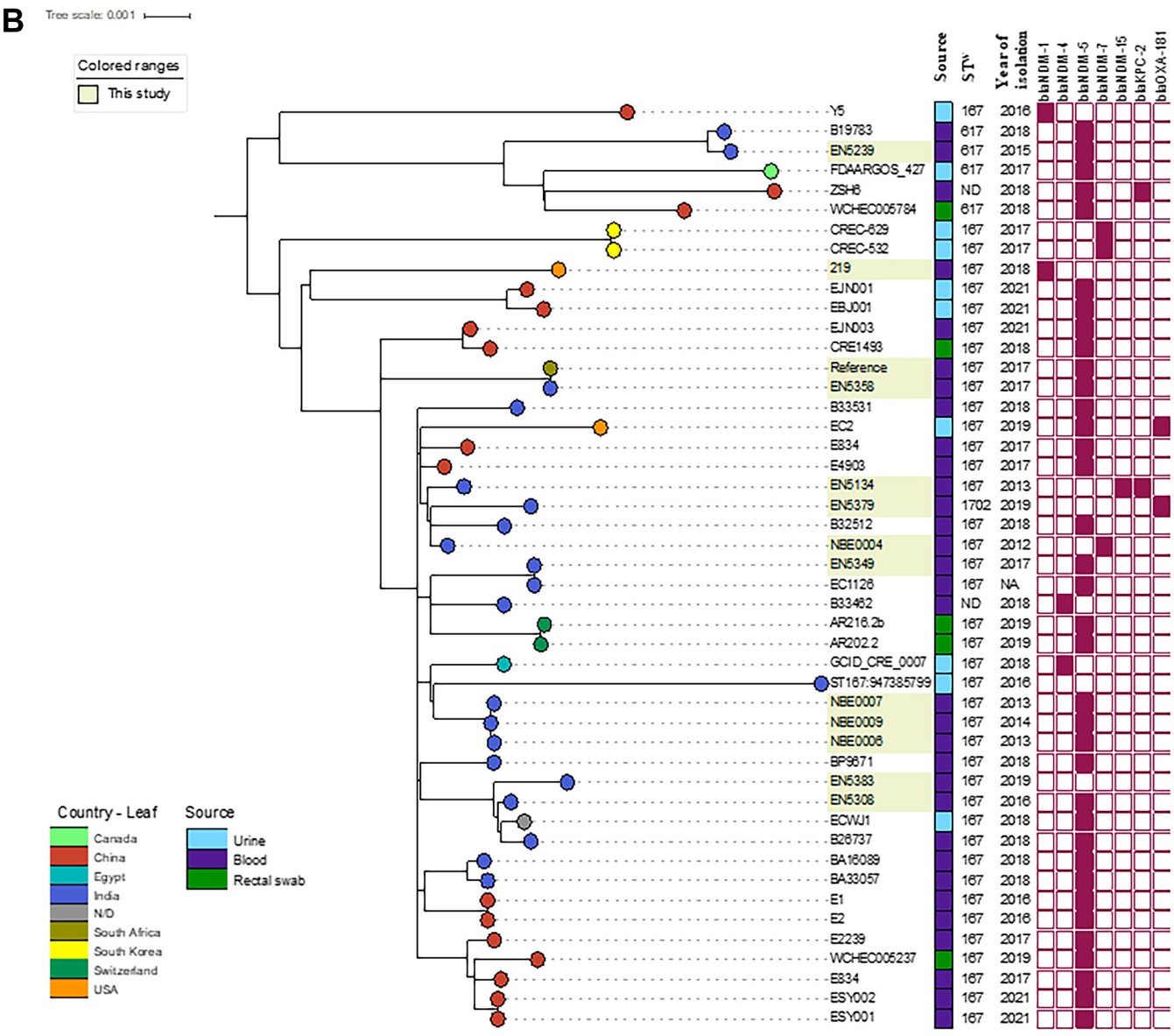

**FIG 4** (Continued)

neonates. The possible relationship between mortality and sepsis due to a $bla_{NDM}^{+ve}$ isolate or due to an epidemic clone was tested (Tables S3A and B). No statistically significant association was noted between mortality and the presence of NDM (Table S3A). A neonate was twice as likely (odds ratio = 2.009615) to have expired when sepsis was due to an epidemic clone than when sepsis was due to a nonepidemic clone (Table S3B). However, the association between mortality and the presence of an epidemic clone was not statistically significant (Pearson $\chi^2_1$ = 2.0724, P value = 0.150). This is possibly due to the small sample size, and, thus, it cannot be interpreted whether the results from Table 4 would hold in the population. An increased neonatal mortality was observed from 2013 onward (before 2013, 39%; after 2013, 57%).

## DISCUSSION

The vulnerability of the neonates and the increasing antimicrobial resistance in causative pathogens lead to morbidity and mortality in this population (1). Although *Klebsiella pneumoniae* and *Acinetobacter baumanii* have lately been noted as the

**TABLE 4** Description of the clinical and bacterial factors along with the outcomes (discharged/expired) of the neonates

| Factors | Discharged | | Expired | | Total | |
|---|---|---|---|---|---|---|
| | *n* | % | *n* | % | *n* | % |
| **Clinical factors** | | | | | | |
| Sex | | | | | | |
| Female | 8 | 22.86 | 14 | 40.00 | 22 | 31.43 |
| Male | 27 | 77.14 | 21 | 60.00 | 48 | 68.57 |
| Gestational age[b] | | | | | | |
| Term | 13 | 37.14 | 10 | 30.30 | 23 | 33.82 |
| Preterm[a] | 22 | 62.86 | 23 | 69.70 | 45 | 66.18 |
| Birth weight[b] | | | | | | |
| Normal birth weight | 9 | 25.71 | 8 | 22.86 | 17 | 24.29 |
| Low birth weight[a] | 26 | 74.29 | 27 | 77.14 | 53 | 75.71 |
| Inborn/Outborn | | | | | | |
| Outborn | 21 | 60.00 | 12 | 34.29 | 33 | 47.14 |
| Inborn | 14 | 40.00 | 23 | 65.71 | 37 | 52.86 |
| Mode of delivery | | | | | | |
| Normal vaginal delivery | 17 | 48.57 | 19 | 59.38 | 36 | 53.73 |
| Caesarean delivery | 18 | 51.43 | 13 | 40.62 | 31 | 46.27 |
| Hospital stay | | | | | | |
| ≤21 days | 14 | 46.67 | 29 | 82.86 | 43 | 66.15 |
| ≥22 days[a] | 16 | 53.33 | 6 | 17.14 | 22 | 33.85 |
| | | | | | | |
| **Bacterial factors** | | | | | | |
| NDM | | | | | | |
| Absent | 22 | 62.86 | 22 | 62.86 | 44 | 62.86 |
| Present | 13 | 37.14 | 13 | 37.14 | 26 | 37.14 |
| Epidemic clone | | | | | | |
| Absent | 19 | 54.29 | 13 | 37.14 | 32 | 45.71 |
| Present | 16 | 45.71 | 22 | 62.86 | 38 | 54.29 |

[a]Low birth weight, <2500 g; Preterm, <37 weeks of gestational age; Prolonged hospital stay, ≥22 days.
[b]Gestational age range, 21 to 41 weeks (median score of 34 weeks); birth weight range, 616 to 3,266 g (median score of 1,630 g).

foremost causes of neonatal sepsis in low-income and middle-income countries (LMICS) (1), *E. coli* still remains relevant (Tables S2A and B).

This study presents an evaluation of *E. coli* causing sepsis over a decade. 74 out of 80 (92.5%) *E. coli* isolates across the years were multidrug-resistant (MDR), and 44% were carbapenem-resistant (all possessing $bla_{NDM}$), indicating that the modification of the WHO guidelines for the treatment of sepsis is long overdue (17). This is particularly important because, although the global rate of under-five mortality is declining rapidly, the decline in the neonatal mortality rate is much slower. In 2020, it was indicated that 47% of all under-five deaths occurred in the newborn period, compared to 40% in 1990. This increase is because the decline in the global level of under-five mortality is happening faster, compared to that of neonatal mortality (22).

This study showed that carbapenem resistance was primarily due to the presence of $bla_{NDM}$, either $bla_{NDM-1}$ (predominantly until 2013) or $bla_{NDM-5/-7}$ (thereafter). The growing predominance of other variants of NDM-1, particularly NDM-5 and NDM-7, has been noted in several studies in India as well as across South East Asia (23, 24). This has also been reflected in sewage specimens that were collected from the northeastern part of India (25). The appearance of variants of NDM had been initially noted in *E. coli*, but they have now been observed in other Enterobacterales, such as *K. pneumoniae* (16, 26). Studies have indicated that the mutation M154L, which is present in both NDM-5 and NDM-7, provides greater adaptability and increased hydrolytic activity over NDM-1 to survive in zinc-deprived conditions (16, 27). This additional feature may provide an evolutionary advantage for the replacement of $bla_{NDM-1}$ with $bla_{NDM-5}$ and other variants in the same clinical setting.

Mobile-genetic elements, such as plasmids, transposons (Tn), and insertion sequences (IS), are the main vehicles for the dissemination of antibiotic resistance genes. The

$bla_{NDM}$ in all of the study isolates were found to be bracketed within a composite transposon (Tn) structure that consisted of IS*Aba125* (either full-length or truncated) at the 5′ end and IS*CR1* (IS91-like element insertion sequence common region 1 family transposase) at the downstream regions. This has also been observed in other $bla_{NDM}$**+ve** isolates (28). IS*CR*1 is often associated with a vast array of antibiotic resistance genes, and it shows a potential for mobilizing adjacent antibiotic resistance genes (28, 29). As both IS*Aba125* and IS*CR*1 were capable of transfer, it is difficult to predict which has contributed to the spread of $bla_{NDM}$ (28). $bla_{NDM}$ was found in transmissible plasmids for all isolates, and most isolates successfully transferred $bla_{NDM}$ via conjugation. Several plasmid scaffolds were identified, some of which were broad host range (IncA/C, L/M, HI, I1, N, and W), and others that were narrow (IncFII, FIA, FIB, and IncX3). The IncX3 plasmids that were identified here were found in 5 out of 22 isolates carrying $bla_{NDM-5/-7}$ and in other previous studies in humans, animals, and the environment (30). The association of the diverse plasmid scaffold is a testament to the rapid and extensive spread of this gene. Several other resistance determinants, such as $bla_{CTX-M}$, $bla_{TEM}$, $bla_{OXA-1}$, and *rmtB* were also transferred along with $bla_{NDM}$, as has been noted in previous studies (31, 32).

The ExPECs in the study were distributed among 6 phylogroups (A, B1, B2, C, D, and F) and 27 ST$^{IP}$s. This indicates that the majority of infections or acquisitions were independent and that cross-transmission in the unit was limited to specific cases. This unit caters to high-risk deliveries and out-born (born in a different hospital and referred to this unit) admissions, and this probably accounts for the diversity of the strains that was noted.

A large proportion of the septicaemic isolates belonged to phylogroup A (25%), and $bla_{NDM}$ was predominantly present in these isolates, notably within the epidemic clone ST$^W$167 isolates (33). Studies from Italy, China, India, USA, Egypt, and Finland made similar observations, where ST$^W$167 was associated with NDM-5 (25, 26, 33, 34). In contrast, 34% of the ExPECs belonged to phylogroup B2, in which ST$^W$131 was common, and most of these isolates did not possess $bla_{NDM}$, except for four that harbored $bla_{NDM-1}$ and not $bla_{NDM-5/-7}$. It was also noted that the phylogroup B2 isolates were present across the entire period. However, most isolates, irrespective of their phylogroups, possessed resistance determinants, such as $bla_{TEM}$, $bla_{SHV}$, $bla_{OXA-1}$, *qnrB*, *aac-(6′)-Ib* and *aac-(6′)-Ib-cr*, and, importantly, $bla_{CTX-M-15}$. This indicates that although both $bla_{NDM-1}$ and $bla_{CTX-M-15}$ are plasmid mediated, their acquisition remains different. $bla_{NDM}$ is primarily associated with ST$^W$167, whereas $bla_{CTX-M-15}$ is associated with both ST$^W$167 and ST$^W$131.

Several studies have shown that commensal *E. coli* are the members of phylogroups A, B1, and C (8). The proportion of septicemia due to these three phylogroups was 50% (A+B1+C) in this study. This high proportion of infection due to commensals could be due to the translocation of these isolates through the immature gut barrier or due to the immature immune system in the sick neonates (35). This was also previously noted in premature neonates with low-birth weights (36) as well as in immunocompromised patients (37). Previous studies analyzing adult septicemia cases showed lower proportions of phylogroups A, B1, and C (20%), in contrast to the results of this study (23) (Table S2B).

Most studies evaluating blood isolates reported a high proportion of B2 (and, to a lesser extent, D), and this was also noted in this study (B2 + D = 45%). Most of the B2 isolates that were studied belonged to ST$^W$131 (subgroup I) with serotypes O25:H4 and subclade H30Rx. Other B2 subgroups, including III (ST$^W$c127), IV (ST$^W$c141), VI (ST$^W$c12), and VII (ST$^W$c14) were also isolated here and are known to be prevalent among ExPEC strains (34).

This study showed that neonates infected with an epidemic clone had a higher risk of mortality (odds ratio = 2.009615). Considering this, the two predominant epidemic clones ST$^W$167 and ST$^W$131 were further analyzed. The genomic relatedness of the ST$^W$167 and ST$^W$131 study isolates were compared, both internally and with a selection of global isolates available in NCBI. As most of the ST$^W$167 strains in this study possessed $bla_{NDM}$, these

genomes were compared to other $bla_{NDM}$+**ve** global ST$^W$167 strains. Most of the study ST$^W$167s (except for a few that were within 15 pairwise SNPs) were distinct and possessed few VFs. The serotypes of the ST$^W$167 study strains were O101:H9 with an *rfb* locus and *fimH* was absent. In contrast to the results of our study, a recent study focusing on the genomics of ST$^W$167 reported serotype O89b:H4 possessing *Klebsiella* sp. specific K48 capsular type (mucoid colony morphology) (33). O89b:H4 strains have also been reported elsewhere (23, 33). In spite of the difference in serotypes, the study isolates had plasmid type F36:A10:B, which is similar to those of the O89b:H4 strains. This reiterates the previously made observation that ST$^W$167 can acquire different capsular types, which is a hallmark for other epidemic clones, such as *E. coli* ST$^W$131 (33).

In the case of ST$^W$131, the basis of selection of global strains for comparison with the study strains was irrespective of the presence of NDM, as $bla_{NDM}$ was not frequent in these strains. Most previous studies have noted that the ST$^W$131 isolates were homogenous (34, 38). However, a SNP analysis of the genomes revealed that our study strains were heterogeneous. This variability could be due to the accessory genome or the variable gene pool, which primarily consisted of genes of unknown functions, transposable and prophage elements, and plasmid-mediated genes (39). Further work analyzing the ST$^W$131 chromosome from this study against ST$^W$131 is warranted but was outside the scope of this study. A collection of ExPECs from different continents reported that there was a convergence of resistance and virulence in ST$^W$131 *papGII*+ lineages and that typical ARGs (*aadA5*, *dfrA12*, *mdf (A)*, *mph(A)*, *sul1*, *tetA*, *aac-6'-Ib-cr*, and $bla_{CTX-M-15}$) were associated with this lineage, as observed here (40). Although several resistance determinants were present in the study ST$^W$131 isolates, $bla_{NDM}$ was rarely found. Apart from the differences in the prevalence of $bla_{NDM}$ in ST$^W$167 (12/14) and ST$^W$131 (4/15), the prevalence of VFs also differed in ST$^W$167 and ST$^W$131. Tourret et al. indicated that extraintestinal virulence resulted from the additive effects of VFs, meaning that the higher numbers of VFs in the B2 strains attest to their disease-causing potential (39).

**Conclusions.** This longitudinal study focusing on neonatal bloodstream infection shows clear differences in the sequence types and serotypes between carbapenem-resistant (primarily $bla_{NDM}$) and susceptible isolates. The comparison of genomes, including the study and global strains, reveals the diversity of ExPECs. The accessory genome plays an important role in antibiotic resistance and also in the diversity of the strain causing sepsis. It is difficult to demarcate commensal and pathogenic *E. coli*, as an equal proportion of strains from both groups caused sepsis. The increasing numbers of carbapenem-resistant high-risk clones is worrisome. As the pool of whole-genomes continues to increase, our understanding of ExPECs will be enriched, and both classical and molecular studies remain important tools in sepsis and antimicrobial resistance research.

## MATERIALS AND METHODS

**Collection and identification of *E. coli*.** Bacterial isolates were primarily isolated from the blood of septicaemic neonates (newborns within 28 days of birth) over a period of approximately 10 years (2009 to 2019) from a tertiary care center, namely, the IPGME&R and SSKM hospital of Kolkata in India. Isolates were identified as *E. coli* and preserved in 20% glycerol at −80°C. Due to some unforeseen circumstances, few isolates could be collected during 2012. For all of the other years, all of the *E. coli* from blood were included. In addition, a collection of *E. coli* isolated from other neonatal specimens *viz.* endotracheal aspirate (ET), peritoneal fluid (PF), pus, and stool, from 2011 to 2014 were also analyzed.

*E. coli* isolates were identified via biochemical tests (2009 to 2017) and by using a Vitek 2 compact system (2018 to 2019), using a card Vitek 2 GN (bioMérieux SA, Marcy l'Etoile, France).

**Antibiotic susceptibility, MIC of meropenem, and resistance determinants.** The susceptibility to different antibiotics, such as piperacillin (100 $\mu$g), cefoxitin (30 $\mu$g), cefotaxime (30 $\mu$g), ciprofloxacin (30 $\mu$g), trimethoprim-sulfamethoxazole (1.25 mg/23.75 $\mu$g), aztreonam (30 $\mu$g), amikacin (30 $\mu$g), gentamicin (10 $\mu$g), meropenem (30 $\mu$g), colistin (10 $\mu$g), and tigecycline (15 $\mu$g) (BD Diagnostics, Franklin Lakes, NJ, USA) was quantified via a conventional disk diffusion assay for the isolates that were collected from 2009 to 2017 (41) and by using a Vitek 2 AST 280 from 2018 onward. Results were interpreted following standard guidelines (42). The MIC for meropenem (MEM) was determined for each isolate via the microbroth dilution method (MBD) (43).

PCR was carried out for the detection of carbapenemases ($bla_{VIM, IMP, SPM-1, GIM-1, SIM-1, KPC, SME, SPM, NDM, GES, OXA-48 like}$) (31, 43), $\beta$-lactamases ($bla_{SHV, TEM, OXA-1, CTX-M}$) (31, 44), AmpCs ($bla_{MOX, CMY, DHA, ACC, MIR/ACT, FOX}$) (45), 16S rRNA methylase-encoding genes (*rmtA*, *rmtB*, *rmtC*, *rmtD*, and *armA*) (46), and plasmid-mediated quinolone resistance (PMQR) genes (*qnrA*, *qnrB*, *qnrS*, *qnrC*, *qnrD*, *aac(6')-Ib-cr*, *qepA*, *oqxA*, *oqxB*) (31, 47).

**Sequencing of carbapenemases.** Amplicons of $bla_{NDM}$ and $bla_{OXA-48}$ were subjected to Sanger sequencing in an automated DNA sequencer (Applied Biosystems, DNA Analyzer, PerkinElmer, USA) and were aligned with GenBank reference gene sequences that are available in NCBI (http://www.ncbi.nlm .nih.gov/genbank) (31). The results of the Sanger sequences were later validated with the whole-genome data.

**Mating-out assay.** The transfer of the $bla_{NDM}$-harboring plasmid (donor) to sodium azide-resistant *E. coli* J53 (recipient) was performed in a solid-state mating assay using a medium supplemented with MEM (2 mg/L) and sodium azide (100 mg/L). For some isolates, conjugation was not successful. Hence, purified plasmid DNA was electrotransformed using electrocompetent *E. coli* DH10B cells (Invitrogen, CA, USA) in a Gene Pulser II (Bio-Rad Laboratories, Hercules, CA, USA). Transformants (TFs) were screened in the presence of MEM (2 mg/L). The presence of $bla_{NDM}$ was confirmed, along with other $\beta$-lactamase genes, via PCR in transconjugants (TCs) and TFs (32). The MIC for MEM was determined for the TCs and TFs via MBD assay (43).

**Plasmid typing.** The plasmid replicon typing of the $bla_{NDM}{}^{+ve}$ wild-type isolates (WT), along with their TCs and TFs, was determined via PCR-based replicon typing (PBRT) assay using a PBRT Kit (DIATHEVA, Italy) (48, 49). In addition, IncF typing (pMLST) was performed by analyzing WGS data (https://cge.food.dtu.dk/ services/pMLST/).

**Genetic environment and integron analysis of $bla_{NDM}{}^{+ve}$ isolates.** The genetic environment of $bla_{NDM}$ was determined via PCR using two sets of primers (32). The PCR data were verified with WGS data, and the upstream and downstream regions of $bla_{NDM}$ in the isolates were determined.

For the integron analysis, the integrase type and variable regions of 5′ to 3′ conserved sequences were determined from the WGS data. The data were submitted to the INTEGRALL site database (http:// integrall.bio.ua.pt) for nomenclature (32).

**Pulsed-field gel electrophoresis (PFGE).** The clonal relatedness among the $bla_{NDM}{}^{+ve}$ isolates was determined via PFGE, applying the standardized pulseNet procedure (http://www.cdc.gov/pulsenet/ protocols.htm). The similarity between the isolates was generated based on the Dice correlation coefficient. The unweighted pair group method with arithmetic mean (UPGMA) was selected for a cluster analysis with 1.5% tolerance and optimization. Isolates showing at least 95% similarity were considered to be identical (50).

**Whole-genome sequencing (WGS).** Based on the results of the PFGE, all of the distinct $bla_{NDM}{}^{+ve}$ isolates and one representative isolate from each clonal cluster were subjected to WGS. Furthermore, isolates of ST131 (Warwick scheme) ($bla_{NDM}{}^{+ve}$ and $bla_{NDM}{}^{-ve}$) were also prepared for WGS. Genomic DNA was extracted by using a Wizard Genomic DNA Purification Kit (Promega) for the short read paired-end sequencing (2 × 150 cycles). The library preparation was carried out using a Nextera XT Kit (Illumina Inc., San Diego, CA). The WGS was executed on an Illumina platform (Illumina Inc., San Diego, CA), using an Illumina NovaSeq 6000 sequencer. The sequences were assembled and processed for further analysis (Supplementary Text).

Using the short read data, the following online-based services were achieved: (i) phylogroup by Clermont typing (http://clermontyping.iame-research.center/); (ii) MLST for sequence type identification (https://cge.food .dtu.dk/services/MLST/); (iii) SeroTypeFinder (https://cge.food.dtu.dk/services/SerotypeFinder/) for serotype identification; (iv) VirulenceFinder and VFDB for virulence gene identification (https://cge.food.dtu.dk/services/ VirulenceFinder/); (v) FimTyper for fimH allele determination (https://cge.food.dtu.dk/services/FimTyper/); (vi) CHTyper (https://cge.food.dtu.dk/services/CHTyper/) for C-H type determination; (vii) ResFinder (https://cge .food.dtu.dk/services/ResFinder/) and Resistance Gene Identifier (https://card.mcmaster.ca/analyze/rgi) for resistance gene identification; (viii) PlasmidFinder for the detection of plasmid replicon types (https://cge.food .dtu.dk/services/PlasmidFinder/); (ix) the integral site for the nomenclature of the integron sequences (http:// integrall.bio.ua.pt); and (x) ISfinder for the identification of IS elements (https://isfinder.biotoul.fr/). Where applicable, all of the PCR data were corroborated with the genome data.

Two phylogenies were performed, based on single nucleotide polymorphisms (SNP). ST$^W$131 and ST$^W$167 ($bla_{NDM}{}^{+ve}$) study isolates with relevant *E. coli* genomes were collected from NCBI, following a literature search in PubMed (Supplementary Text). IQtree (v2.0) was used to generate a SNP phylogeny, and iTOL (v6.0) was used to annotate and visualize the phylogenetic tree. Additionally, a core genome phylogenetic tree was generated for all of the study $bla_{NDM}{}^{+ve}$ isolates using Panaroo (v.1.2.8) and IQtree (v2.0).

**Phylogroups, sequence types (ST), and virulence factors (VFs).** A multiplex PCR assay was used to broadly classify the isolates into one of the *E. coli* phylogenetic groups (A, B1, B2, C, D, E, or F). The discrimination of phylogroups A or C and D or E were further carried out by using the C and E specific primers in a singlex PCR assay (51, 52).

For a more precise characterization of the isolates, multilocus sequence typing (MLST) was performed using the Institut Pasteur MLST scheme (Paris, France) (https://pubmlst.org/bigsdb?db=pubmlst _mlst_seqdef) (53). The clonal relatedness of the isolates was established using the global optimal eBURST (goeBURST) algorithm for clustering closely related STs under a single clonal complex (ST$_C$) (54). The STs of the Institut Pasteur scheme (ST$^{IP}$) were converted to the equivalent ST$_C$s of the Warwick scheme (ST$^W$) (51) (Olivier Clermont personal data).

19 different VFs were screened for all of the isolates (55), and they were categorized as follows: (i) adhesion/fimbriae (afimbrial adhesion [*afa*], type 1 fimbriae [*fimH*], P fimbriae [*papA-C-Gll*], type S or F1C fimbriae [*sfa/foc*]); (ii) toxins (cytolethal distending toxin [*cdtB*], cytotoxic necrotizing factor 1 [*cnf1*], haemolysin A [*hlyA*]); (iii) siderophores (aerobactin [*iucC*], iron acquisition [*iroN*$_{E.coli}$], yersiniabactin receptor [*fyuA*]); (iv) invasins (invasion of brain endothelium factor [*ibeA*]); (v) protectins and serum resistance (serum resistance-associated outer membrane protein [*traT*], colicin [*cvaC*], iron-regulated gene homologue

adhesin [*iha*], uropathogenic-specific protein [*usp*]); and (vi) miscellaneous (pathogenicity associated island marker [PAI], type II capsule [MTII]) (55, 56).

**Statistical analysis.** Neonatal mortality and related clinical and bacterial factors from a sample of 70 observations (fewer observations for some factors due to missing data) were studied. This was done by comparing the proportions of patients in different groups. For some bacterial factors, a bivariate analysis was carried out to test associations with neonatal mortality. The Chi-square test of association was conducted where appropriate. All of the data analyses were executed using Stata version 17.0.

**Ethics approval and consent to participate.** The samples were collected as a part of the routine diagnosis of sepsis. Hence, a waiver of informed consent was granted. The study protocol was reviewed and approved by the Institutional Ethics Committee of ICMR-National Institute of Cholera and Enteric Diseases (No. A-1/2019-IEC, dated October 18, 2019). Patient records and information were anonymized and deidentified prior to analysis.

**Data availability.** All of the genome data were submitted to the NCBI database under the BioProjects PRJNA548120 (EN5134), PRJEB33565 (EN5356 and EN5358), and PRJNA801430 (the rest of the isolates).

## SUPPLEMENTAL MATERIAL

Supplemental material is available online only.
**SUPPLEMENTAL FILE 1**, TIF file, 9 MB.
**SUPPLEMENTAL FILE 2**, TIF file, 5.6 MB.
**SUPPLEMENTAL FILE 3**, DOCX file, 0.03 MB.

## ACKNOWLEDGMENTS

We would like to express our gratitude to G.A. Jacoby and S. Brisse for providing the PCR controls and to Thomas Jove (INTEGRALL) for helping with the curation of the integron sequences. We thank the team of curators of the Institut Pasteur MLST and whole-genome MLST databases for curating the data and making them publicly available at http://bigsdb .pasteur.fr/. We sincerely thank all of the staff and clinicians of the Department of Neonatology, SSKM, and IPGME&R hospital for their cooperation in handling and sending the samples as well as Molay Kuity for the laboratory assistance and data maintenance.

This study was supported by the Indian Council of Medical Research (ICMR) intramural fund. A.B. and S.M. were supported by fellowships from the ICMR.

We declare that we have no competing interests.

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
