## [Reviewer comments · Microbiology Spectrum]

Microbiology Spectrum

A Decade-long Evaluation of Neonatal Septicaemic *Escherichia coli*: Clonal Lineages, Genomes and New Delhi Metallo- β -Lactamase Variants

Amrita Bhattacharjee, Kirsty Sands, Shravani Mitra, Ritojeet Basu, Bijan Saha, Olivier Clermont, Shanta Dutta, and Sulagna Basu

Corresponding Author(s): Sulagna Basu, National Institute of Cholera and Enteric Diseases

Review Timeline:

Submission Date:	December 21, 2022
Editorial Decision:	February 13, 2023
Revision Received:	May 20, 2023
Accepted:	May 27, 2023

Editor: Krisztina Papp-Wallace

Reviewer(s): Disclosure of reviewer identity is with reference to reviewer comments included in decision letter(s). The following individuals involved in review of your submission have agreed to reveal their identity: Jumpei Saito (Reviewer #1)

Transaction Report:

DOI: <https://doi.org/10.1128/spectrum.05215-22>

February 13, 2023

Dr. Sulagna Basu
National Institute of Cholera and Enteric Diseases
Bacteriology
P33, CIT Road, Scheme XM, Beliaghata
Kolkata, West Bengal 700010
India

Re: Spectrum05215-22 (**A Decade-long Evaluation of Neonatal Septicaemic *Escherichia coli*: Clonal Lineages, Genomes and New Delhi Metallo- β -Lactamase Variants**)

Dear Dr. Sulagna Basu:

Link Not Available

Sincerely,

Krisztina Papp-Wallace

Journals Department
Reviewer comments:

Reviewer #1 (Comments for the Author):

The fact that a long-term study of ExPECs was conducted on newborns admitted to the NICU is valuable information. Although the study was conducted in a single region, the content of the study was substantial, and I believe it will contribute to future research activities. On the other hand, I cannot find any evidence that the understanding of the transition of resistant strains itself will change the current practice of antimicrobial chemotherapy. In order to understand that the emergence of resistant strains is a critical situation, it is necessary to analyze the relationship between detected bacteria or isolates and outcomes (e.g., death, prolonged hospital stay, etc.). I have attached a comment on how. Please refer to the reference.

Executive Summary.

There is no information on the source of the isolates considered in this study (e.g., information about the population, geographic region, etc.).

L93 "This versatility is observed in its genome and also in its function or potential to cause disease" A reference to this statement is needed.

L106 What is "[WHO]"?

L171 "Whole genome sequencing (WGS)" WGS is spelled out in the sentence. The description of the method is redundant. It is too long compared to other items and should be shortened or moved to a supplement file.

Result.

L272 "Seventy E. coli were identified from the blood of septicaemic neonates (2009 to 2019)."

The number of cases, patient background, and outcomes (treatment outcomes) are unknown. Also, whether the results are similar to other parts of India or unique to this facility needs to be added to the discussion.

L231 "Isolates were resistant to second (52/80, 65%) and third-generation cephalosporins (67/80, 84%)" In the Methods section, the study on cephalosporins I am aware that the study regarding cephalosporins is not explicitly mentioned in the Methods section.

L267 "Transmissibility of metallo- β -lactamases and plasmid profile:" Is a colon necessary?

Discussion

L434 "under-5 mortality" The phrase "under-five" is also mentioned.

L480-489 "Several studies have shown that commensal E. coli are " The sentence is redundant due to the extensive use of literature citations. I think that you should state shortly and precisely what you want to assert from the results of this review. Overall, we believe that the authors over-explain the contents of the cited references.

Table 1. Although the table is organized in an easy-to-read manner, please consider creating a table that includes previous reports and results from other regions to determine if the results are appropriately compared or discussed.

Table 2. It is unclear what the criteria are for the type of rules.

The description of the discussion items is polite and clear, but is it possible to separate the items to make them more readable?

The reviewer has studied several reports of genetic testing of bacterial isolates, and it is not particularly rare to find such a heterogeneous population of isolates in a single institution's study?

Reviewer #2 (Comments for the Author):

Would be interesting to know if clinical outcomes changed after 2013 when transition from NDM-1 to NDM -5/ -7.

Staff Comments:

Preparing Revision Guidelines

Please return the manuscript within 60 days; if you cannot complete the modification within this time period, please contact me. If you do not wish to modify the manuscript and prefer to submit it to another journal, please notify me of your decision immediately so that the manuscript may be formally withdrawn from consideration by Microbiology Spectrum.

Reply to Editor's comments

Reviewer #1

The fact that a long-term study of ExPECs was conducted on newborns admitted to the NICU is valuable information. Although the study was conducted in a single region, the content of the study was substantial, and I believe it will contribute to future research activities. On the other hand, I cannot find any evidence that the understanding of the transition of resistant strains itself will change the current practice of antimicrobial chemotherapy. In order to understand that the emergence of resistant strains is a critical situation, it is necessary to analyze the relationship between detected bacteria or isolates and outcomes (e.g., death, prolonged hospital stay, etc.). I have attached a comment on how. Please refer to the reference.

1. Executive Summary

There is no information on the source of the isolates considered in this study (e.g., information about the population, geographic region, etc.).

Answer: Isolates were collected primarily from the blood of septicaemic neonates admitted to a tertiary care hospital (IPGME&R and SSKM hospital of Kolkata, India). This tertiary care hospital in Kolkata, West Bengal caters to a population in Kolkata and also patients from different districts (at a distance of 100 km radius from Kolkata) of West Bengal.

2. L93 "This versatility is observed in its genome and also in its function or potential to cause disease" A reference to this statement is needed.

Answer: Authors have included the reference in reference number 9.

3. L106 What is "[WHO]"?

Answer:

Authors have expanded the term "WHO" as World Health Organization on **Page No. 6 & Line No. 104 in the Basu_S_Marked Up Manuscript.**

4. L171 "Whole genome sequencing (WGS)" WGS is spelled out in the sentence. The description of the method is redundant. It is too long compared to other items and should be shortened or moved to a supplement file.

Answer:

This is the sub-heading for the following section, hence it is not spelled out.

The authors have shortened this section in the revised manuscript (**Basu_S_Marked-Up Manuscript**) and moved part of it to the **Supplementary file** under the "**Method for whole genome sequence (WGS) analysis**" subheading.

5. Result.

L272 "Seventy *E. coli* were identified from the blood of septicaemic neonates (2009 to 2019)." The number of cases, patient background, and outcomes (treatment outcomes) are unknown. Also, whether the results are similar to other parts of India or unique to this facility needs to be added to the discussion.

Answer: A separate table (**Table 4**) depicting the association of outcome related to clinical and bacterial factors of neonates have been presented in the manuscript.

Two supplementary tables (**Supplementary Table S2A, S2B**) highlighting studies on *E. coli* causing neonatal sepsis from different parts of India and across the globe have been incorporated. Authors have compared data from these studies wherever applicable.

6. L231 "Isolates were resistant to second (52/80, 65%) and third-generation cephalosporins (67/80, 84%)" In the Methods section, the study on cephalosporins I am aware that the study regarding cephalosporins is not explicitly mentioned in the Methods section.

Answer: Antibiotic susceptibility pattern was determined for cephalosporins (cefoxitin, cefuroxime as 2nd gen and cefotaxime, ceftriaxone as 3rd gen cephalosporins) as a part of the antibiotic susceptibility test [disk diffusion data of cefoxitin, cefotaxime (2009-2017) and cefuroxime, ceftriaxone in VITEK®2 AST 280 (2018-2019)]. This has been clarified in the foot note of **Table 1 (Page No. 32 & Line No. 759)**.

7. L267 "Transmissibility of metallo- β -lactamases and plasmid profile:" Is a colon necessary?

Answer: Authors have now omitted the colon.

8. Discussion

L434 "under-5 mortality" The phrase "under-five" is also mentioned.

Answer: Authors have rephrased the term "under-5 mortality" by "under-five mortality" on **Page No. 20 & Line No 444-445**.

9. L480-489 "Several studies have shown that commensal *E. coli* are " The sentence is redundant due to the extensive use of literature citations. I think that you should state shortly and precisely what you want to assert from the results of this review. Overall, we believe that the authors over-explain the contents of the cited references.

Answer: Authors have modified the part of the discussion, taking into consideration comments from the reviewer.

10. Table 1. Although the table is organized in an easy-to-read manner, please consider creating a table that includes previous reports and results from other regions to determine if the results are appropriately compared or discussed.

Answer: Authors have included supplementary Table S2A and S2B in supplementary file representing list of published literature from different regions of India (S2A) and other countries (S2B) focusing on neonatal sepsis.

11. Table 2. It is unclear what the criteria are for the type of rules.

Answer: Authors have now modified Table 2.

12. The description of the discussion items is polite and clear, but is it possible to separate the items to make them more readable?

Answer: Authors have now modified the discussion.

13. The reviewer has studied several reports of genetic testing of bacterial isolates, and it is not particularly rare to find such a heterogeneous population of isolates in a single institution's study?

Answer: Yes, the authors agree to this comment. However, since this unit had a heterogeneous population of isolates, hence it is mentioned in the study.

14. Reviewer #2:

Would be interesting to know if clinical outcomes changed after 2013 when transition from NDM-1 to NDM -5/ -7.

Answer: A descriptive table with patient data and outcomes has been incorporated (**Table 4**) in the main manuscript. The possible relationship between mortality and sepsis due to a *bla*_{NDM}^{+ve} isolate was tested (**Supplementary Table S3B**) but not found to be statistically significant. It was noted that mortality increased from 2013 onwards (before 2013-39%, after 2013-57%). This has been incorporated on **Page No. 19 & Line No 432-433**.

Note:

New sequence types have been obtained for two study isolates which have been included in Table 1, Table 2, Figure 1 and Supplementary Figure 1.

May 27, 2023

Dr. Sulagna Basu
National Institute of Cholera and Enteric Diseases
Bacteriology
P33, CIT Road, Scheme XM, Beliaghata
Kolkata, West Bengal 700010
India

Re: Spectrum05215-22R1 (**A Decade-long Evaluation of Neonatal Septicaemic *Escherichia coli*: Clonal Lineages, Genomes and New Delhi Metallo- β -Lactamase Variants**)

Dear Dr. Sulagna Basu:

Your manuscript has been accepted, and I am forwarding it to the ASM Journals Department for publication. You will be notified when your proofs are ready to be viewed.

Sincerely,

Krisztina Papp-Wallace
Editor, Microbiology Spectrum
